# Synthetic neural-like computing in microbial consortia for pattern recognition

Ximing Li [1], Luna Rizik [1], Valeriia Kravchik[1], Maria Khoury[1], Netanel Korin [1] & Ramez Daniel [1✉]

Complex biological systems in nature comprise cells that act collectively to solve sophisticated tasks. Synthetic biological systems, in contrast, are designed for specific tasks, following computational principles including logic gates and analog design. Yet such approaches cannot be easily adapted for multiple tasks in biological contexts. Alternatively, artificial neural networks, comprised of flexible interactions for computation, support adaptive designs and are adopted for diverse applications. Here, motivated by the structural similarity between artificial neural networks and cellular networks, we implement neural-like computing in bacteria consortia for recognizing patterns. Specifically, receiver bacteria collectively interact with sender bacteria for decision-making through quorum sensing. Input patterns formed by chemical inducers activate senders to produce signaling molecules at varying levels. These levels, which act as weights, are programmed by tuning the sender promoter strength Furthermore, a gradient descent based algorithm that enables weights optimization was developed. Weights were experimentally examined for recognizing $3 \times 3$-bit pattern.

[1] Department of Biomedical Engineering Technion—Israel Institute of Technology, Technion City, Haifa, Israel. ✉email: ramizda@bm.technion.ac.il

Living systems are constantly engaged in computational processes such as signal detection, processing, and decision making to perform sophisticated tasks. For example, in the vertebrate adaptive immune system, the invasion of pathogens triggers a series of actions from multiple cell types to protect the organisms[1]. The computational properties of biological systems can emerge from coordinative and collective interactions among basic components[2,3]. These components can be neurons interacting with other cells in the brain, bacteria communicating with other members in a community, or receptors participating in signaling pathways[4].

In contrast to natural living systems, synthetic biocircuits excel at only a narrow range of human-defined computations[2,3]. One reason is that they are designed following principles from computer engineering, represented by implementations such as toggle switches[5], oscillators[6], memory devices[7], and state machines[8]. A few major computational paradigms have been widely adopted for circuit design, namely digital design, analog design, and control design[9–12]. Digital design takes inputs of binary-coded levels, highlighting concepts such as logic gates and Boolean functions. Analog design and control design handle a range of continuous input levels, focusing on system stability and design dynamics. Nevertheless, depending on design paradigms, synthetic gene circuits face challenges such as host limitations, random fluctuations, and unwanted interactions with host cells[13,14].

Some studies have exploited the dynamic structure of cell communities and obtained more sophisticated functions than in individual cells[15,16]. Multi-cellular systems naturally allow distributed and parallel computing. Using these features, studies have successfully implemented edge detection[17] and spatial pattern formation[18,19]. Furthermore, cells in communities can be organized with flexibility, such as being layered for logic gates[20,21] or arranged to form various ecosystems[3].

So far, synthetic biocircuits are often designed for specific tasks and cannot easily be adapted for solving a wide range of problems. However, as the synthetic biology community attempts to create "smart cells" for a variety of applications[22], it is important to build circuits that can be adapted and optimized without explicit programming. Multi-cellular systems provide a solution to this issue. In these systems, computations can naturally emerge from flexibly interconnected cells that act concertedly. The flexibility and interconnection, similar to the structure in neural networks, offer a novel design that can be adapted to solve a range of problems, overcoming the lack of generality in mainstream paradigms. Inspired by biological neural networks, artificial neural networks (ANNs) are adaptive computing models that are commonly adopted to solve a wide range of tasks[23]. In this study, we draw an analogy from inter-cellular relations to ANNs and demonstrate that ANN provides a powerful design to engineer multi-cellular systems. ANNs model the network structure with layers of connected units. The connecting strengths between units, namely the weights, can be trained to achieve specific tasks. This trainable feature allows ANNs to "learn" the weight values, so that tasks involving decision-making, such as pattern recognition, can be learned. A simple unit of ANN is perceptron (Fig. 1a), which performs a weighted summation of inputs, and becomes activated for decisions. Despite the simplicity, perceptrons can classify patterns that are linearly separable, which means input points on a plane belonging to distinct categories can be geometrically separated by a line (or a hyper-plane for patterns in high dimensions). By modifying weights, a perceptron unit can be used for pattern recognition.

Specifically, here we explore the collective behaviors in *Escherichia coli* (*E. coli*) bacteria cell consortia, demonstrating that the interactions between cell groups can be used to implement a perceptron network for pattern classification (Fig. 1b). The patterns are represented by the amount of inducer OC6 (acyl-homoserine lactone 3OC6-HSL) described in binary levels, either with or without inducer. The inducer can activate a group of sender bacteria to produce QS signaling molecules OHC14 (acyl-homoserine lactone 3OHC14:1-HSL), which diffuse into and activate the receivers. The receivers provide the activation function to convert the weighted sum of OCH14 collectively produced from senders into different activation states for classifying input patterns. We vary the strength of the $P_{lux}$ promoter in senders to obtain different weights. We first experimentally examined the QS system using simple 4-bit patterns. Then we developed an algorithm based on gradient descent, which is widely used in artificial intelligence, to optimize weights so that more sophisticated patterns could be categorized. Using our algorithm, we obtained the weights for $3 \times 3$-bit patterns and experimentally tested the weights using the QS system for these patterns. Our implementation demonstrates a framework to train genetic circuits in silico and satisfy the target functions in vivo for pattern recognition. This implementation provides a prototype to implement neural-network-like computing in living bacteria.

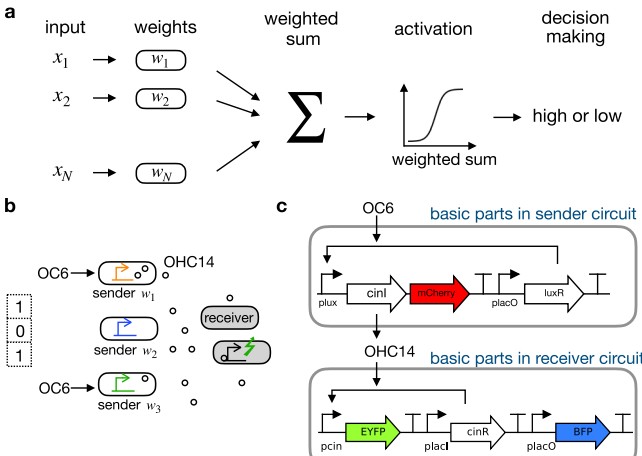

**Fig. 1 Perceptron diagram and genetic network implementation in experiments. a** Schematic representation of a perceptron with an N-elements input. The elements of the input vector $\vec{x}$ are multiplied with the elements of the weight vector $\vec{w}$, giving rise to a weighted sum. The weighted sum acts as input to an activation function, the output of which is taken for decision-making. **b** Illustration of a perceptron network based on quorum sensing between sender and receiver bacteria. An example pattern "101" is represented by high acyl-homoserine lactone 3OC6-HSL (OC6), no OC6 and high OC6. Senders containing promoters with different strengths are shown in different colors, corresponding to different weights. Small circles are signaling molecules (i.e., acyl-homoserine lactone 3OHC14:1-HSL, OHC14) produced by senders. The molecules diffuse across the cell membrane and activate receivers to generate fluorescence signals. **c** Basic genetic circuit parts in senders and receivers. In senders, inducer OC6 binds with constitutively expressed transcription factor LuxR and activates $P_{lux}$ to express mCherry signal and acyl-homoserine-lactone synthase. The latter protein is encoded by *cinI* and catalyzes the synthesis of OHC14, which is the diffusible signaling molecule. In receivers, the OHC14 entering from media binds with transcription factor CinR to form a complex, which activates $P_{cin}$ to express the fluorescence signal enhanced yellow fluorescent protein (EYFP). A constitutively expressed blue fluorescent protein (BFP) signal serves as a control.

## Results

**Circuit engineering in senders and receivers.** The basic parts for senders and receivers are shown in Fig. 1c. In senders, when OC6

is given as an input, the constitutively expressed LuxR binds with OC6 to activate $P_{lux}$, allowing *cinI* and *mCherry* to express. Expression of *cinI* catalyzes the synthesis of OHC14, which diffuses into receivers across the media. In receivers, OHC14 binds with the constitutively expressed CinR transcription factor, forming a complex that activates the $P_{cin}$ promoter. The activity of the $P_{cin}$ is measured by enhanced yellow fluorescent protein (EYFP) signals. However, when mixing senders exposed to different inputs, i.e., with or without OC6, a cross-talk can occur between these senders (Supplementary Fig. 13). Specifically, senders that are initially inactivated ("0") may be affected by the residual OC6 from those that are activated ("1"). To minimize such a cross-talk, we added a double inversion circuit ($P_{roD}$-*tetR* and $P_{tetO}$-*lacI*) to the sender circuit (Fig. 2a). The additional circuit is designed to control the activity of $P_{lacO}$ using anhydrotetracycline (aTc) to affect the expression of LuxR and the activity of $P_{lux}$. In the double inversion circuit, TetR is constitutively produced to repress $P_{tetO}$ and block the LacI expression. Therefore, without aTc, LacI is not expressed and $P_{lacO}$ remains constitutively on to express LuxR. In the presence of aTc, TetR binds aTc and releases from $P_{tetO}$, allowing LacI to express. In this case, LacI represses $P_{lacO}$ and blocks the LuxR expression. Therefore, the presence of OC6 in the cells cannot affect the expression levels of *cinI* and *mCherry*. In Supplementary Notes, we added a discussion part on cross-talk, as well as its impact on analog computations and the experimental protocols.

In addition, we also examined an alternative implementation of senders using PF regulation (Supplementary Fig. 10a). Compared with senders in OL (Fig. 2e), senders with PF exhibit transfer functions that are less homogenously characterized, but are steeper and with higher thresholds. This characteristic allows senders not much affected by residual OC6 when they are mixed and require no aTc dependent double inversion circuit to reduce cross-talks. Moreover, we implemented three-well fluidic devices (Supplementary Fig. 7), which allow spatial separation of cells. In this design, the wells are connected by a channel. Sender bacteria are placed in the two wells on the side and receiver bacteria locate in the center well. OC6 are selectively added to the side wells, according to particular patterns. The device effectively acts as an "AND" classifier when equal weights of wild-type $P_{lux}$ promoters in senders are used in both side wells (Supplementary Fig. 8). Further discussion is provided in Supplementary Notes section "Alternative implementations and Fluidics experiments".

To improve the output dynamic range of the double inversion circuit, we added decoy binding sites (Fig. 2a). In particular, we implemented six decoy lacO operators ($P_{lacO}$ array) to sequester extra LacI produced from $P_{tetO}$[24]. Specifically, the $P_{lacO}$ that expresses LuxR is on a low-copy-number plasmid (LCP), and the decoy $P_{lacO}$ array is on a medium-copy-number plasmid (MCP). The sequestering effect can adjust the output range of the double inversion circuit. As shown in Fig. 2b, with the $P_{lacO}$ array on MCP, aTc (20 ng ml⁻¹) reduces green fluorescent protein (GFP) expression (i.e., the activity of $P_{lacO}$ on LCP) by approximately 10 times (Fig. 2b, green bars). In contrast, without the $P_{lacO}$ array (Fig. 2b, orange bars), GFP expression is constantly low regardless of aTc levels.

The purpose of the activation function in perceptron is to nonlinearly map the weighted summation onto separate states for decision making (Fig. 2c). Therefore, the steeper the activation function, the sharper the decision boundary, which is easier for the network to make decisions. In order to improve the steepness of the activation function, we implemented a positive feedback regulation in receivers, which sharpens the transfer function of receiver circuits (Fig. 2d). We fitted the transfer functions of the two circuits using a Hill equation, in the form $f(x) = \beta_m \frac{(x/K_d)^n}{1+(x/K_d)^n} + \beta_m \beta_0$. Here $x$ represents the inducer

**Table 1 Fittings parameters using Hill equations for transfer functions of receiver and sender circuits.**

|            | $K_d$     | $\beta_0$   | $\beta_m$  | $n$   |
|------------|-----------|-------------|------------|-------|
| R_PF       | 0.28 μM   | 3.95E−02    | 2.25E+04   | 1.97  |
| R_OL       | 3.41 μM   | 7.23E−03    | 2.57E+04   | 0.809 |
| S_mut40    | 15 μM     | 4.95E−03    | 5521       | 0.352 |
| S_mut7     | 50 μM     | 9.1E−02     | 1906       | 0.482 |
| S_mut8     | 50 μM     | 1.91E−01    | 729        | 0.54  |
| S_mut15    | 50 μM     | 4.45E−02    | 2679       | 0.426 |
| S_plux_rep | 0.252 μM  | 3.41E−01    | 735        | 0.669 |
| R_mCh      | 1103      | 1.95E−02    | 2.16E04    | 2.33  |

R_PF: the transfer function of receiver circuits with positive feedback (Fig. 2d).
R_OL: the transfer function of receiver circuits with open loop (Fig. 2d).
Row S_mut40 - S_mut15: transfer functions of activator senders with mutated $P_{lux}$ (Fig. 2e).
Row S_plux_rep: transfer functions of $P_{lux}$ repressor senders (Fig. 2e).
Row R_mCh: transfer function of receiver circuits with horizontal axis as mCherry levels (Fig. 4c).

amount, $K_d$ is the dissociation constant between the inducer-TF complex and the promoter, $\beta_m$ is the maximal activity of the promoter, $\beta_m \beta_0$ describes the basal activity of the promoter, and $n$ is the Hill coefficient, representing the cooperativity involved in TF-inducer binding and TF-promoter binding. As shown in Table 1, the positive feedback (PF) circuit exhibits a higher hill coefficient ($n = 1.97$ in row R_PF) than the open-loop (OL) circuit ($n = 0.809$ in row R_OL). We provided further details on fitting transfer functions in Supplementary Notes section "Transfer function fitting".

The perceptron weights can be implemented by varying the promoter strength to affect the transfer function of sender circuits in response to OC6. To obtain various perceptron weights, we mutated the first four base pairs in $P_{lux}$ and obtained promoters with a range of almost continuous varying strengths (Supplementary Fig. 3). We selected four distinct strength levels for the perceptron network (Fig. 2e). These mutated senders all exhibit upward transfer functions as OC6 increases and comprise various positive weights in the perceptron network. To build more sophisticated functions, we also implemented a weight with a negative sign. For example, a XOR logic function can be implemented with two-perceptron layers consisting of negative and positive weights[25]. Here, we used the $P_{lux}$ repressor promoter to engineer the negative weight by placing the LuxR binding site downstream to a strong constitutive promoter (Fig. 2e). The arrangement allows binding of the OC6–LuxR complex to the promoter to block RNA Polymerase from initiating the transcription. The transfer functions of sender circuits containing $P_{lux}$ activators and the $P_{lux}$ repressor were experimentally measured. The $P_{lux}$ activators were fitted using the same equation as described previously for receiver circuits. Notice that $K_d$, $n$ and $\beta_0$ are similar for different mutations, leaving $\beta_m$ as the defining variable for the transfer functions. Therefore, we factor out $\beta_m$ and consider it as the weight variable. The $P_{lux}$ repressor was fitted using an equation in the form $f(x) = \alpha_m \frac{1}{1+(x/K_d)^n} + \alpha_0 \alpha_m = \hat{\alpha}_m (\frac{(x/K_d)^n}{1+(x/K_d)^n} - \alpha_0 - 1)$, where $K_d$ and $n$ are the same as previously described. Similar to $\beta_m$ and $\beta_m \beta_0$, $\alpha_m$ and $\alpha_m \alpha_0$ describe the max activity and minimal activity of the $P_{lux}$ repressor promoter. $\hat{\alpha}_m$ is the weight for the repressor promoter and has a negative value, consistent with the negative slope of the transfer function. Aside from transcription repression, a negative weight can also be implemented using quorum quenching enzymes[26]. A large part of these enzymes is lactonases and acylases, which can reduce quorum-sensing signals via degradation. Since these enzymes usually act on a range of lactone signals, circuit designs require careful characterization of the specificities.

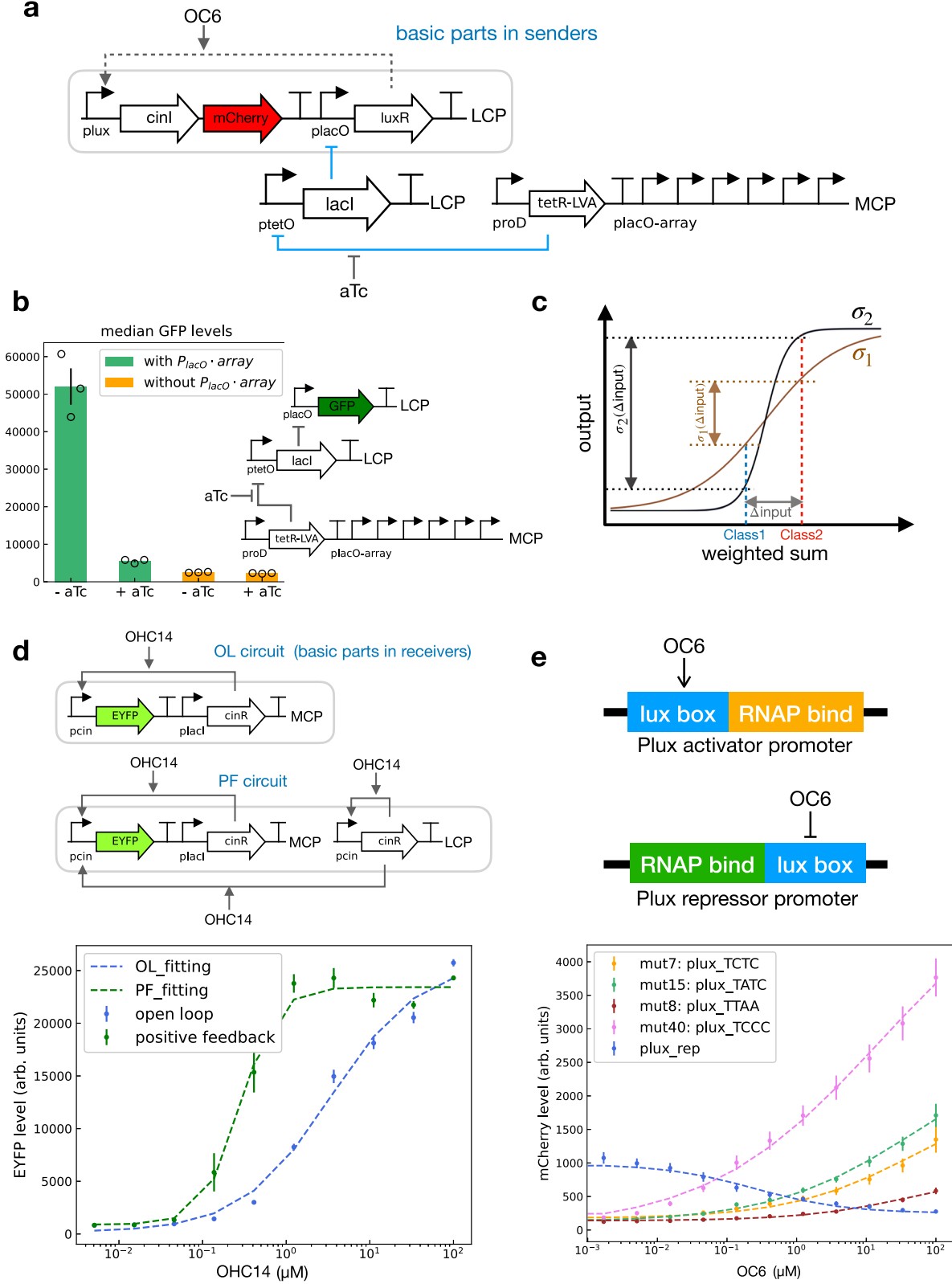

For further explanation on the rationale of implementing negative weight, please see Supplementary Notes section "Negative weight".

**Classification of 4-bit patterns**. With the engineered circuits in senders and receivers, we next examined whether they can be built into a perceptron to classify patterns of OC6. We started

with an arbitrary weight vector, selected the patterns that can be classified by the weights and implemented using previously built genetic circuits. The weight vector [450, 3500, 900, 3500] was selected (Fig. 3a). The values correspond to estimates of mCherry levels measured from mutated senders after 210 min incubation when the OC6 concentration is 33 μM (Fig. 2e). This OC6 level is in the high inducer analog range (33 μM is not high enough to

**Fig. 2 Circuit engineering in senders and receivers. a** Double-inversion parts added to sender circuits to minimize the cross-talk between senders. The double-inversion circuit can be controlled by anhydrotetracycline (aTc) to repress the activity of $P_{lacO}$ on a low-copy-number plasmid (LCP). A $P_{lacO}$-array locate on a medium-copy-number plasmid (MCP) can tune the activity of the sender circuit. **b** The decoy binding sites $P_{lacO}$-array on MCP are necessary to maintain a large output dynamic range in the double inversion circuit. Here the assay was tested using the double inversion circuit shown on the right. With the $P_{lacO}$-array, the expression of green fluorescent protein (GFP) is turned off in the presence of aTc (20 ng ml$^{-1}$) (green bars, +aTc vs −aTc). Without the $P_{lacO}$-array, GFP expression stays low regardless of the aTc concentration (orange bars, +aTc vs −aTc). Data are presented as mean values of median EYFP ± SEM from independent replicates ($n = 3$). Median EYFP of individual replicate is marked in circle. Fluorescence measurement are in arbitrary units (arb. units). **c** Schematic drawing to show how the slope of the activation function affects the decision-making of the perceptron. Both the black curve and the brown curve ($\sigma_1$) are activation functions ($\sigma_2$) mapping the weighted sum to different activation levels. The black curve is steeper than the brown curve. For a pair of inputs from two classes, the steeper the activation function, the larger difference between the output of the two inputs. **d** Genetic circuit diagram for receivers in open loop (OL) and with positive feedback (PF) regulation. The circuit topology affects the receiver transfer function slope. The OL circuit is the same as the basic receiver circuit in (Fig. 1c). In PF circuit, the feedback part locates on LCP. PF and OL result in different transfer functions. Data are presented as mean values of median EYFP ± SEM from independent replicates ($n = 3$). Dashed lines are fittings using Hill equations from experiment measures, with fitting parameters shown in Table 1. Source data are available in the Source data file. **e** Schematic representation of the $P_{lux}$ activator promoter and $P_{lux}$ repressor promoter. Transfer functions of senders with various mutated $P_{lux}$ (TCTC, TATC, TTAA, TCCC) activator promoters and $P_{lux}$ repressor promoter. Data are presented as mean values of median mCherry ± SEM from multiple replicates ($n = 10$ for TCTC, $n = 10$ for TATC, $n = 9$ for TTAA, $n = 11$ for TCCC, $n = 11$ for $P_{lux}$ repressor). Dashed lines are fittings using Hill equations from experiment measures, with fitting parameters shown in Table 1.

saturate the mutated $P_{lux}$ activity, where the max level used in Fig. 2e is 100 μM). There are in total sixteen 4-bit binary patterns, a subset of which can be well classified by the weight vector. In brief, for each pattern, we calculated the product of the weight vector and the pattern, and compared the product value with thresholds for decision making. Patterns with product values larger than an upper threshold are grouped as one class. Likewise, patterns with product values smaller than a lower threshold are grouped as another class. The selected patterns are shown in Fig. 3a. There are four patterns locate within a boundary range, which is estimated as [4000, 4500]. Thus, a perfect classification of all sixteen patterns depends on the threshold and sharpness of the receiver transfer function. We included further explanation regarding the boundary range and design specifics to achieve perfect classification in Supplementary Notes section "Explanation on classifying 4-bit patterns".

Senders grown overnight were diluted 100 times and incubated at 37 °C in separate wells for 210 min (Fig. 3b, further information is provided in "Methods"). For input bits "1", senders were incubated in LB containing OC6. For input bits "0", senders were incubated in LB containing aTc to reduce the unwanted cross-talk between senders after mixing. Next, all senders were diluted 100 times in LB, incubated for 30 min, and mixed together. The sender mixture was then transferred into receiver solutions (50 times diluted from overnight culture) in a 1-to-19 volume ratio and incubated at 37 °C for 90 min. Finally, EYFP and BFP signals from the incubated sample were measured using flow cytometry. In Fig. 3b, the input pattern $\overrightarrow{p}_8$ ([1,0,1,1]) is demonstrated as an example. Note that pattern $\overrightarrow{p}_{10}$ ([1,1,1,0]) also yields the same sender mixture, i.e., a mix of mut8 with OC6, mut40, mut7 with OC6, and mut40 with OC6.

Median EYFP levels from receivers were averaged from three trials for all patterns (Fig. 3c). As expected, receivers are activated to higher levels by patterns in set1 ($\overrightarrow{p}_6$ to $\overrightarrow{p}_{11}$) than those in set0 ($\overrightarrow{p}_0$ to $\overrightarrow{p}_5$). The BFP expression levels are uniform across patterns, indicating that the distinctions among observed EYFP signals are not due to the variability in receiver growth or a disproportionate mix of senders and receivers. These results suggest that OHC14 collectively produced by senders with various $P_{lux}$ mutations (i.e., weights) can be effectively summated in receivers. The activation function of receivers with a positive feedback loop was capable of classify the input patterns.

**Developing an algorithm to learn the weights**. Subsequently, we developed an algorithm to systematically search for $P_{lux}$ promoter strength that can map patterns onto target activation levels in receivers. We modeled the system in a series of steps (Fig. 4a), following the procedures in experiments. First, we started with senders of random weights. Selected senders were induced by chemical inputs, then mixed and incubated together with receivers. Next, the receiver output were compared with predefined target levels for error calculation (i.e., loss function) to update weights. Newly updated weights were then fed into the model again to iterate the steps until the receiver output became sufficiently close to targets. In particular, the activation function $\sigma$ of receivers was experimentally measured (Fig. 4c), using the sender mCherry levels as input.

This iterative approach is a common algorithm used to train ANNs[27] based on the gradient descent method. Intuitively, consider the difference between the model output and target as a mountain ridge with changing slopes (Fig. 4b). The algorithm, like a hiker, aims to find the valley of the ridge, i.e., to minimize error. Starting at a random point, the hiker chooses the direction for the next step based on the slope at that point. Following the downward slope, step by step, the hiker can gradually reach the lowest point. The curve shown in Fig. 4b is simplified when optimizing one parameter. When multiple parameters are optimized, as in the case of a multi-element weight vector, we need to search for the lowest point on a high dimensional "mountain" surface. For a particular point on the surface, there can be multiple downward directions. The algorithm thus follows the steepest downward direction by calculating the gradient of error.

A solution can be obtained following the gradient-based method. Depending on the initial position and step size, the solution can sometimes become trapped in a local minimal point and may not be optimal. In this case, a direct search can be subsequently carried out in an area nearby the sub-optimal solution to search for the optimal weights. Furthermore, the method described so far assumes weights are continuous variables. During experiments, however, we used mutated $P_{lux}$ promoter that exhibit discrete strength levels. As a realistic consideration, we adapted the gradient-based algorithm by constraining weights to discrete values with additional weights update rules. We present the details of the algorithm in Supplementary Notes section "Algorithm development".

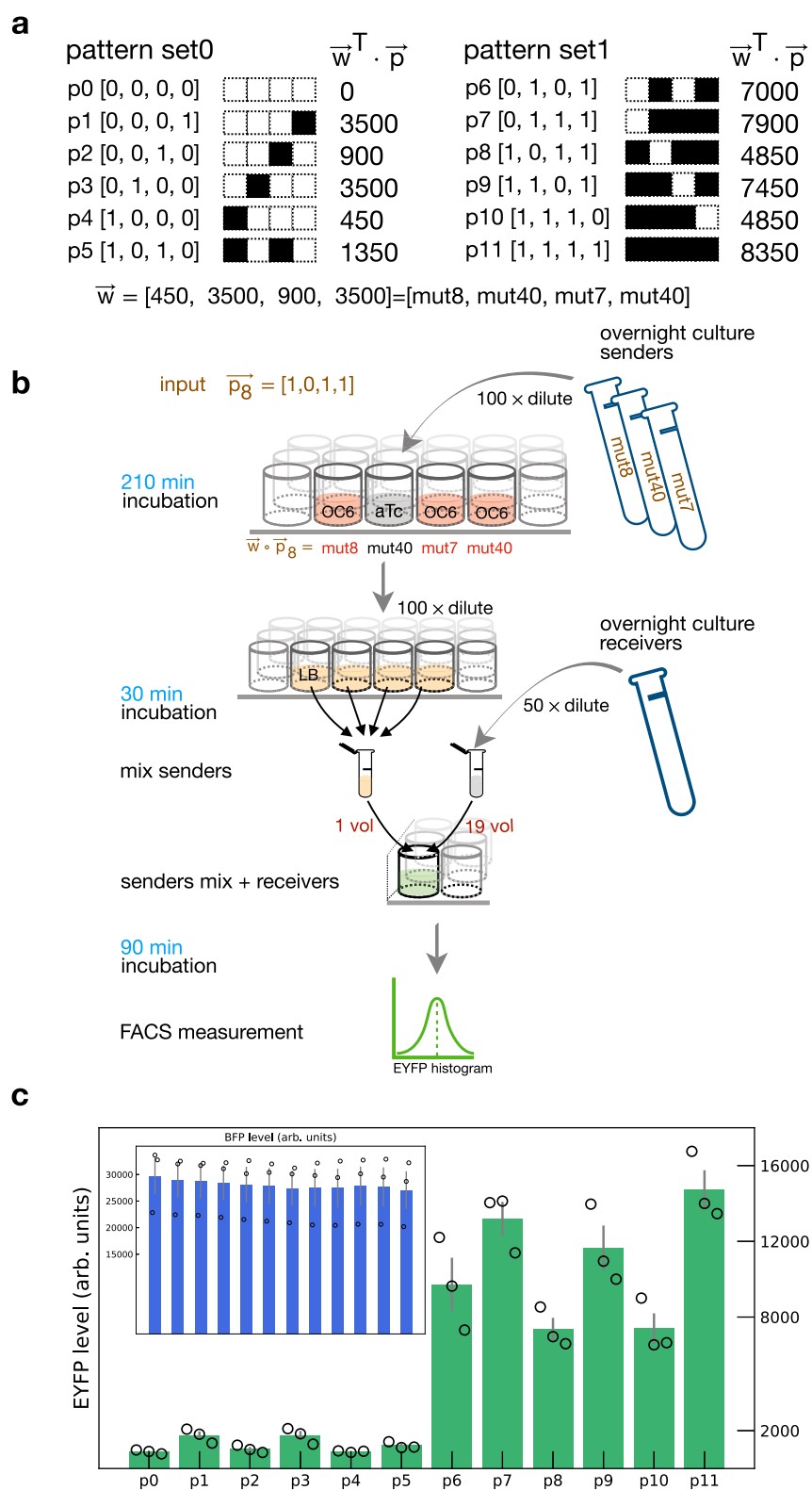

**Fig. 3 Classification of 4-bit patterns. a** Two sets of 4-bit patterns with binary value entries. Weight vector [450, 3500, 900, 3500] is used to separate the patterns. **b** Workflow of experiment procedures using pattern [1,0,1,1] as an example. In brief, senders grown overnight were 100 times diluted and incubated at 37 °C in separate wells for 210 min. Depending on the input bits, senders were incubated in LB containing either OC6 ("1") to control the activity of $P_{lux}$, or aTc ("0") to prevent the cross-talk between senders after mixing. Next, all senders were diluted 100 times in LB, incubated for 30 min, mixed and transferred into receiver culture in a 1-to-19 volume ratio. The sender-receiver mixture was incubated at 37 °C for 90 min and the fluorescence signals were then measured using flow cytometry. **c** Flow cytometry measurement of receiver output for 4-bit patterns. EYFP and BFP (inset) are presented as mean values of median EYFP ± SEM from independent replicates ($n = 3$). Median values of individual replicates are marked in circles. Source data are available in the Source data file.

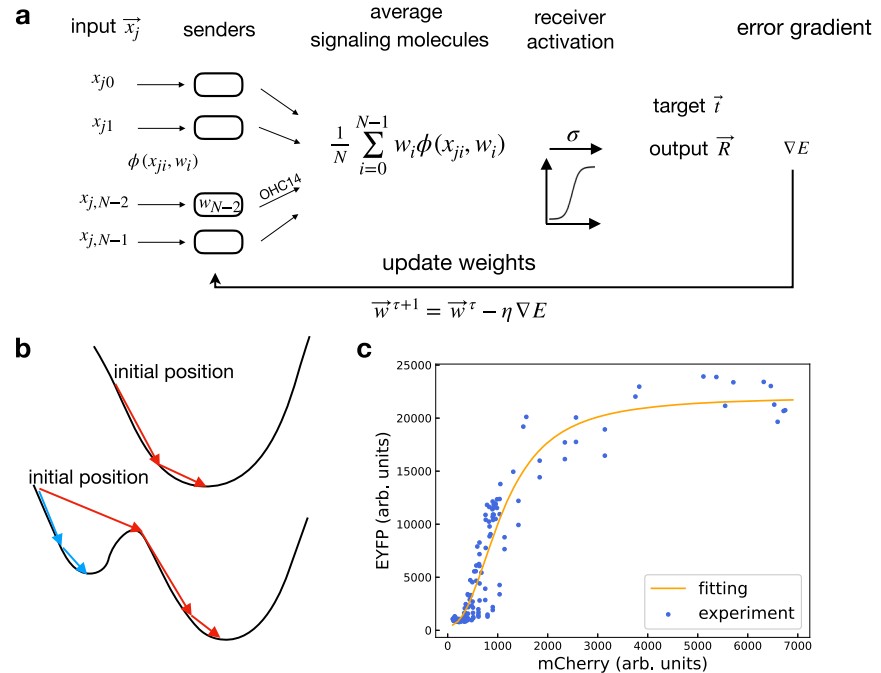

**Fig. 4 Algorithm for learning weights. a** Schematic representation of steps involved in the algorithm. The input $\vec{x}_j$ is one pattern from the complete pattern set. The weight vector $\vec{w}$ represents the promoter strengths of the $P_{lux}$ promoter in sender circuits. The product of the input and weight vectors is the amount of signaling molecules OHC14 collectively produced by senders. After the sender mixture was transferred into receiver culture, the signaling molecules activate the receiver circuits, effectively as an input to the function $\sigma$. The output of $\sigma$, $\vec{R}$, is compared with a target vector $\vec{t}$ to calculate an error gradient, which is used to update weight values for the next iteration. $\eta$ is the learning rate parameter. **b** Cartoons of error function curves illustrate the iterative procedure to search for the minimum point. In the lower graph, red arrows represent the large iterative step size, which leads to the global minimum point. Blue arrows represent the small iterative step size, which leads to the local minima. **c** Receiver EYFP levels as a function of mCherry levels in senders. Data were obtained from 12 sets of mCherry and EYFP measurements. Senders of different $P_{lux}$ mutations are incubated with varying amounts of OC6 and mixed with receivers following the steps in Fig. 3b. Source data are available in the Source data file.

**Classification of $3 \times 3$-bit patterns**. Following our method to learn weights, we performed classification of more complicated pattern sets (Fig. 5a) than the 4-bit patterns, using the genetic circuits constructed previously. We choose three sets of $3 \times 3$-bit patterns that have previously been used for classification with state-of-the art electronic circuits[28]. The pattern sets include three categories, "z", "v", and "n". Each set includes one ideal or noiseless pattern and nine noisy ones with one bit flipped (Fig. 5a). We included three weight vectors ($\vec{w}_0$, $\vec{w}_1$, $\vec{w}_2$) to recognize the three pattern categories. In particular, the $\vec{w}_0$ vector generates an output vector approximate the target vector $\vec{t}_0$ (Fig. 5a). Similarly, the $\vec{w}_1$ and $\vec{w}_2$ vectors can produce output vectors approximate $\vec{t}_1$ and $\vec{t}_2$, respectively.

We obtained the three vectors using our optimization method. The simulated output is shown in Fig. 5b. The three vectors can be obtained by first following the gradient-based method and then a direct search, since the gradient-based approach alone does not necessarily converge to the optimal solution. Alternatively, the vectors can also be achieved by updating weights constrained to a given set of discrete values. Notably, many combinations in the product of weights and inputs are repetitions (Supplementary Fig. 20). Only ten combinations are unique (Fig. 5b). Such a phenomenon is related to the nature of patterns and the performance of experimental procedure. A section is provided in Supplementary Notes on repetitions in patterns for further discussion. Subsequently, we experimentally tested the ten non-repeating combinations using selected weights. As shown in Fig. 5c, the first four products ($\vec{w}_0^T \cdot \vec{p}_0$, $\vec{w}_0^T \cdot \vec{p}_1$, $\vec{w}_0^T \cdot \vec{p}_3$ and

$\vec{w}_0^T \cdot \vec{p}_5$) result in high EYFP levels in receivers and the rest products result in low output. This is consistent with the simulation results in Fig. 5b, as the products of $\vec{w}_0$ and all patterns in "z" ($\vec{p}_0$ to $\vec{p}_9$) are expected to be high. Meanwhile, the BFP expression levels are uniform across all patterns. Furthermore, weights with two distinct levels can also classify the patterns with good performance. The simulated output is provided in Supplementary Fig. 5.

**Simulations of classifying more sophisticated patterns**. Patterns in the real world are often larger than $3 \times 3$ pixels. To understand how the algorithm performs on larger patterns, we further expanded the same pattern sets to $5 \times 5$, $7 \times 7$ and $9 \times 9$ bits. Similar to the $3 \times 3$-bit patterns, each pattern set includes one clean pattern and $N^2$ noisy ones. Based on our algorithm, we can classify binary $5 \times 5$, $7 \times 7$ and $9 \times 9$-bit patterns with good performance, assuming either discrete weights (Fig. 6a) or continous positive weights (Supplementary Fig. 21). Interestingly, the high and low output in $5 \times 5$-bit patterns are more seperated than those in $3 \times 3$-bit patterns. The separation is further improved in $7 \times 7$-bit and $9 \times 9$-bit patterns. This is likely due to the increased pattern sparseness in large-scale patterns. Sparseness is related to the proportion of "0" bits of a pattern. For $3 \times 3$-bit "z", "v" and "n" patterns, the percentage of "0" states is 0.33 (3/9); for $5 \times 5$, $7 \times 7$ and $9 \times 9$-bit patterns, the percentages are 0.64 (16/25), 0.73 (36/49) and 0.79 (64/81), respectively. Theoretical works suggest that capacity of neural network increases for sparser patterns[29].

Circuits based on logic gates are often designed for patterns in binary values. One advantage of ANN over logic gate design is

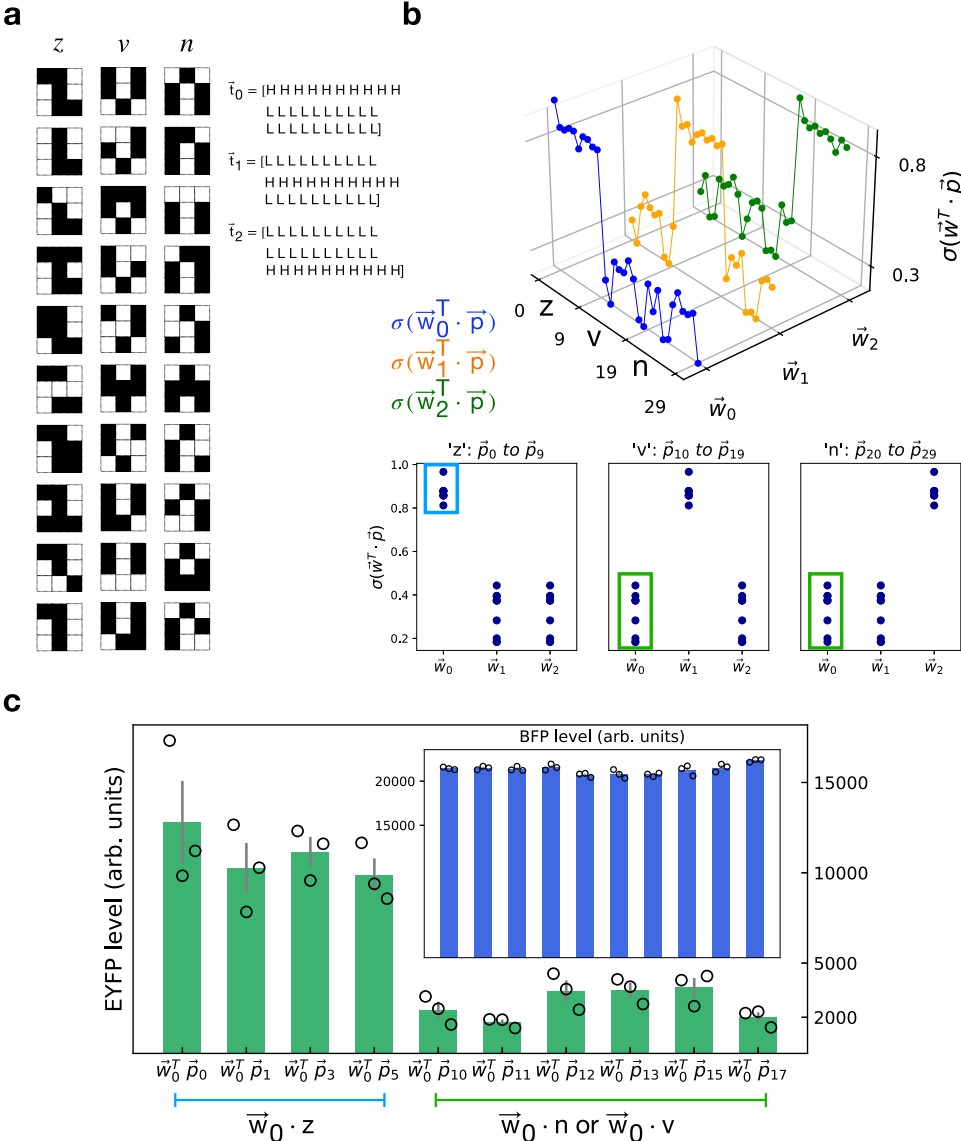

**Fig. 5 Classification of 3 × 3-bit patterns. a** 3 × 3-bit patterns for "z", "v", and "n". Each pattern category consists of one ideal pattern and nine noisy patterns. Each noisy pattern has one-bit flip from the ideal pattern. Three target vectors are shown on the right, in which "H" represents high target values and "L" represents low values. **b** Simulated receiver output presented in 3D. The weights were obtained using the gradient-based method. Three output graphs are shown separately for three weight vectors in distinct colors. Each colored curve consists of 30 dots, corresponding to the 30 patterns (10 patterns × 3 categories), in the order of "z" (patterns 0 to 9), "v" (patterns 10 to 19), and "n" patterns (20 to 29). The shape of each curve is consistent with each target vector. Below is the receiver output for each pattern category. These sub-figures are equivalent to rotating the 3D graph leftward and viewing from the side. For "z" patterns (the left sub-figure), $\vec{w}_0$ gives rise to high output, whereas $\vec{w}_1$ and $\vec{w}_2$ result in low output. Similarly, $\vec{w}_1$ results in high output for "v" patterns (the middle sub-figure) and $\vec{w}_2$ leads to high output for "n" patterns (the right sub-figure). Dots are overlapped. It is worth noting that there are only ten distinct output values. For example, the high state dots marked in a blue square in the left sub-figure have the same values as the dots in the middle and right sub-figures. **c** Flow cytometry measurement of receiver output for 3 × 3-bit patterns. In this case, nine groups of senders were selected, corresponding to the nine bits in each pattern and nine elements in weight vectors. Senders of different groups were incubated separately and mixed with receivers. The fluorescence signals from receivers were measured. EYFP and BFP (inset) are presented as mean values of median EYFP ± SEM from independent replicates ($n = 3$). Median values of individual replicates are marked in circles. Not all products of patterns and weights are presented. As discussed in the text, many pattern-weight products are repetitions (Supplementary Fig. 20). We presented only the unique and non-repeating ones in experiments. The first four bars represent products of $\vec{w}_0$ and "z" patterns, which are marked in a blue square in Fig. 5b left sub-figure. The rest bars represent $\vec{w}_0 \cdot \vec{v}$, equivalently $\vec{w}_0 \cdot \vec{n}$, which are marked in green squares in Fig. 5b middle and right sub-figures, respectively. Source data are available in the Source data file.

that ANN also works on patterns with graded values. To see whether our design can handle non-binary patterns, we modified the same 5 × 5, 7 × 7 and 9 × 9-bit patterns by varying the "1" elements to random values from 0.5 to 1 (Fig. 6b). Simulated classification results indicate that our algorithm can achieve good performance with non-binary patterns using either discrete

weight levels (Fig. 6b) or continous positive weights (Supplementary Fig. 21). In particular, compared with binary patterns, more weight levels are necessary to achieve good classifications for non-binary patterns (Supplementary Fig. 22). We list the relevant weight values that are obtained using the algorithm in the Supplementary Tables 5–7.

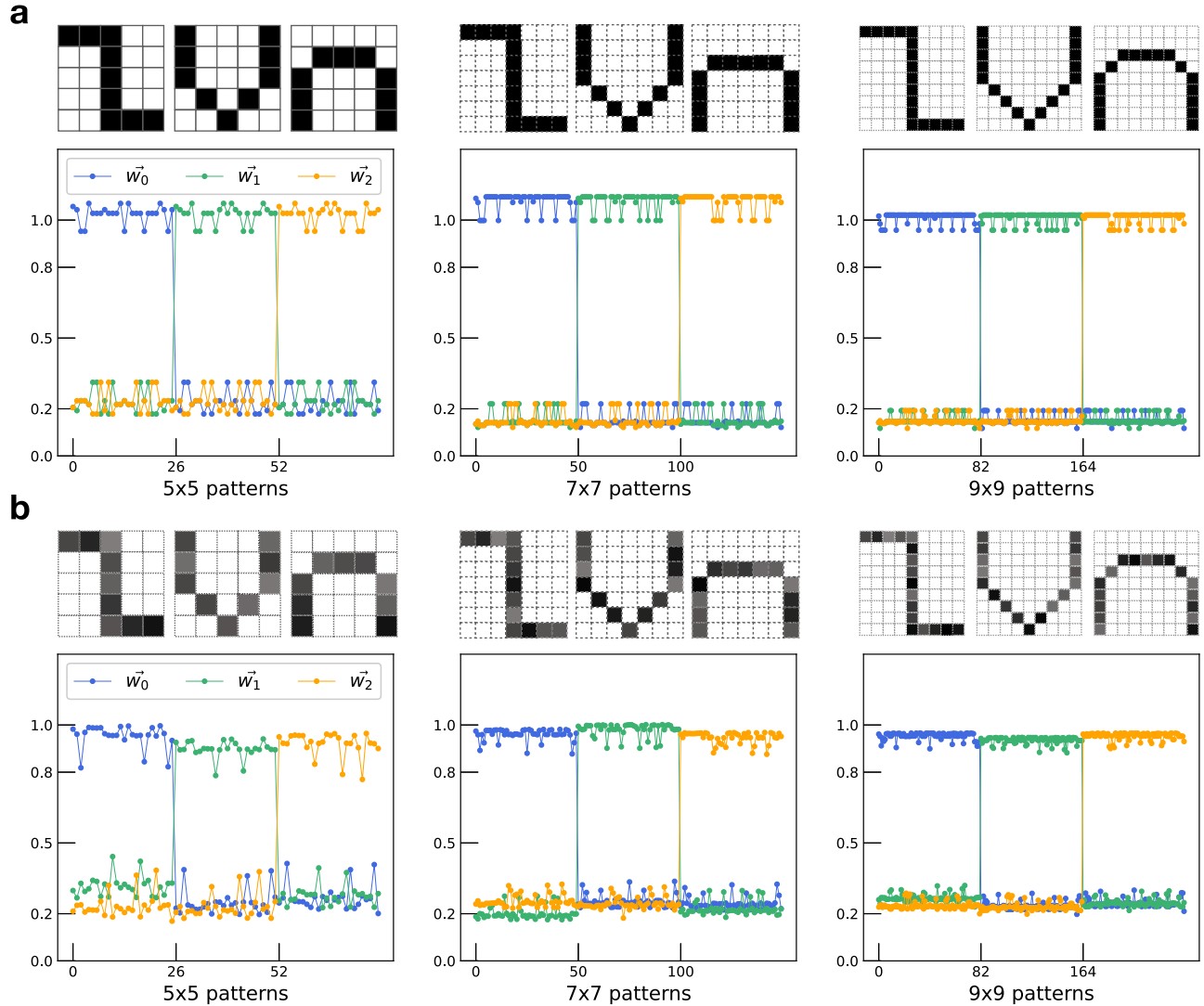

**Fig. 6 Simulated results for classifying more sophisticated patterns. a** Simulated receiver output for $5 \times 5$, $7 \times 7$, and $9 \times 9$-bit patterns. In the lower panel, receiver outputs for patterns in three categories are overlaid together. The outputs are marked by lines in different colors and are presented similarly as in Fig. 5b. For the $5 \times 5$ set, there are in total $(1 + 25) \times 3 = 78$ patterns; for the $7 \times 7$ set, the total number is $(1 + 49) \times 3 = 150$; and for the $9 \times 9$ set, $(1 + 81) \times 3 = 246$. Only the noiseless patterns are shown. **b** The same as in (**a**) but for patterns with non-binary inputs values. To generate these patterns, each element in a pattern is multiplied by a random number between 0.5 and 1 to emulate the randomized OC6 input. In particular, the same random number vector (of size $1 \times 25$) is used for all 78 $5 \times 5$ patterns. Similarly, one random vector (of size $1 \times 49$) for all 150 $7 \times 7$ patterns and one random vector (of size $1 \times 81$) for all 246 $9 \times 9$ patterns. Weight values are listed in Supplementary Tables 5–7.

## Discussion

We demonstrated a genetic circuit implementation of perceptron networks based on bacterial quorum sensing. The implementation allows *E. coli* to recognize chemical input patterns. Along with the genetic circuits, we also formulated an algorithm to obtain appropriate weights for pattern classification. We experimentally tested the implementation first using 4-bit patterns and then $3 \times 3$-bit patterns. In both experiments, the patterns were successfully classified. We further showed that the algorithm can solve more sophisticated patterns, which are larger in size and having non-binary inputs. The simulated results demonstrate the capability of the algorithm to tackle a range of pattern sets. The algorithm-guided approach provides an example that couples in silico design and in vivo implementation.

Spatial patterns have been extensively studied in ANN literature for developing computer vision algorithms. In the present study, we focused on classifying spatial patterns using genetic circuits. Reactions for individual bits or pixels are incubated

separately before being mixed for summation. Aside from the protocol as shown in Fig. 3b, the experiment procedure can be carried out in fluidic devices, which have compartments for individual reactions and channels for signaling molecules to diffuse. Such devices can reduce the manual steps involved in the process. As a demonstration, we implemented three-well devices (Supplementary Figs. 7–9), allowing for the classification of 2-bit patterns. Furthermore, our architecture can also be adapted to classify non-spatial patterns, in which case each pixel or bit represents distinct chemical or biomolecule species. Using molecular parts sensitive to only one specific inducer, we can reduce the cross-talk between different inducers. We demonstrated an example of classifying 2-bit patterns coded for inducers OC6 and Arabinose (Supplementary Fig. 1).

It is worthwhile pointing out that the fluorescence measurement can vary from trial to trial depending on the quality of growth media and antibiotics. Thus, it is not possible to match the exact weight values as calculated using the algorithm. More

importantly, through the experiments, we demonstrated that decision making in living cells does not rely on exact behaviors. On the other hand, the noise is inherent to genetic circuits and can aggregate during the summation of multiple inputs. Hypothetically, when reaching a certain number of inputs, the receivers cannot necessarily distinguish the on-off ratio in one input bit versus noise fluctuation. Thus, noise limits the maximum number of inputs that a network can distinguish. Also, when cascading multiple layers of circuits, noise propagation across layers can limit the operation of networks[30,31]. Further discussion regarding multi-layer networks is provided in Supplementary Notes section "Capacity of perceptron and scale-up to multi-layer networks".

Previous works have implemented classifiers in living cells or cell-free systems. Logic circuits have been implemented to detect cancer cells via intra-cellular miRNA markers[32] and to recognize cell surface receptors using engineered DNA aptamers reaction networks[33]. Didovyk et al.[34] implemented a distributed classifier, in which the decision is made at a population scale. Pandi et al.[35] demonstrated a metabolic perceptron to classify 4-bit patterns in cell-free systems. Different from these works, our design is based on communications between two groups of cells with one cell group explicitly make decisions. This architecture demonstrates the power of multi-cellular computing. The two cell groups, i.e., senders and receivers, encapsulate distinct computational functions (e.g., weighting and summations) in separate compartments. In this way, cellular functions are separated in modules and metabolic burdens are distributed across species, allowing for increased system flexibility and scalability. With the design, computational behaviors arise from the interactions between senders and receivers, even though the tasks performed in individual cell groups are relatively primitive. Similar themes have also been presented in other studies for majority sensing[36] and tunable population dynamics[37]. In addition, we implemented weights by varying the relevant promoter strength. Specifically, we customized a repressor promoter for negative weight, which is crucial to classify certain patterns (Supplementary Fig. 19). This feature is not available for implementations using substrate concentrations as weights, in which case weights are non-negative values.

ANNs are the enabling tool for today's artificial intelligence technology. In computer engineering, the capability of learning in ANNs offers a great advantage over traditional combinatorial logic circuits. Recently, ANNs have been implemented alternatively in memristors[28] (i.e., a new form of electronics), optics[38], DNA strands[39], and cell-free systems[35]. Likewise, as a computing model, ANNs could also provide a design architecture in synthetic biology to engineer biological systems with more adaptivity. For example, the framework and algorithm in our study can be used to facilitate the design of living therapeutics, such as targeted drug release system based on engineered probiotic bacteria system[40]. Our proposed system can also be potentially extended to engineer inter-cellular communications in yeasts cells[41] and mammalian cells[19]. For the latter in particular, engineering how tissue cells contact each other would enable new applications for programming tissue development, growth, and repair. In addition, our framework provides a prototype to implement more sophisticated computations based on collective activities from cell groups. Apart from perceptron, there are more complicated network structures that enable the representation of sensory information, which can be investigated to develop advanced functions in the future.

## Methods

**Chemicals**. All chemicals used in the study are of the highest analytical grade. OC6 was obtained from Sigma Aldrich. OHC14 (N-(3-hydroxy-7-cis tetradecenoyl)-L-

homoserine lactone) was obtained from Cayman Chemical Company. Anhydrotetracycline (aTc) was from Takara Bio.

**Strains**. *E. coli* 10β was used for plasmid construction and all experiment assays. All liquid media used in the study was Luria-Bertani-Miller (LB). Two types of antibiotics were used, kanamycin (30 μg ml⁻¹) and cloramphenicol (25 μg ml⁻¹). The specifics of *E. coli* 10β include: araD139 D (ara-leu) 7697 fhuA lacX74 galK (W80 D (lacZ) M15) mcrA galU recA1 endA1 nupG rpsL (StrR) D (mrr-hsdRMS-mcrBC).

**Plasmids construction**. The plasmid pAJM.1642 containing the essential parts in receivers ($P_{cin}$-EYFP and $P_{lacI}$-cinR) was obtained from Christopher Voigt's Laboratory (Addgene plasmid #108535; http://n2t.net/addgene:108535;RRID: Addgene_108535)[42]. All other plasmids were constructed using basic molecular cloning methods[43], including standard steps like PCR, restriction digestion, ligation, and transformation. PCR was carried out in a BioRad S1000 Thermal Cycler. Oligonucleotide primers were synthesized by Integrated DNA Technologies (Coralville, IA). Restriction digestion enzymes were purchased from New England Biolabs (Beverly, MA) and Thermo Scientific FastDigest. Ligations were performed using T4 DNA ligase with ligation buffer from New England Biolabs. For transformation, we used standard heat shock in *E. coli* 10β, followed by colony PCR screening on the next day. Selected colonies were grown overnight for miniprep (BioBasic) and sent for standard sequencing (Macrogen Europe, The Netherlands) using appropriate primers (Supplementary Table S0). Mutations in the $P_{lux}$ promoter were performed using site-directed mutation (Agilent QuickChange lightening), following the manufacture's protocol. Mutations were first performed in a simple circuit, $P_{lux}$-GFP-$P_{lacO}$-luxR. Transfer functions of mutated colonies were characterized. The mutations with desired characteristics were then selected for sequencing and integrated with the other parts in senders.

**Experiments using FACS**. Raw data from FACS presented in the work contain 10,000 events and the abort rate was kept less than 2%. Raw data were preprocessed using density-based gating by a Python library FlowCal[44] (v1.2.2) to obtain 8000 events. All circuit diagrams were drawn using a Python package DNAplotlib[45] (v1.0).

*Cross-talk experiment (Fig. 2b)*. Relevant plasmids used in the experiments were XL267+XL208 (with $P_{lacO}$-array), XL267+LR206 (without $P_{lacO}$-array) (see Supplementary Table 1 for complete list). *E. coli* colonies from transformation plates were inoculated in 4 ml LB solution with 4 μl kanamycin (30 μg ml⁻¹) and 4 μl chloramphenicol (25 μg ml⁻¹), grown overnight at 37 °C in a shaking incubator (Shel Labs SSI5) at 250 rpm. On the next day, overnight cultures were diluted 100 times in LB antibiotics with or without aTc (final concentration 20 ng ml⁻¹). Aliquots (200 μl) of diluted culture were transferred into 96-well plates and incubated at 37 °C for 210 min in a microplate shaker (Lumitron) at 500 rpm.

*Receivers characterization experiment (Fig. 2d)*. Relevant plasmids used in the experiments were XL340+XL291 (positive feedback) and pAJM.1642+LR191 (open loop). *E. coli* colonies were grown overnight at 37 °C in 4 ml LB solution with the appropriate antibiotic combinations. Three colonies were grown for each circuit. Overnight cultures were diluted 50 times and aliquoted into a 96-well plate. Inducer OHC14 (Cayman Chemical Company) was added (final max concentration 100 μM) and then serial diluted three times across wells for each colony. The 96-well plate was taken to a microplate shaker (Lumitron) and incubated for 90 min at 500 rpm.

*Senders characterization experiment (Fig. 2e)*. Relevant plasmids used were XL140mut40+XL208, XL140mut15+XL208, XL140mut8+XL208, XL140mut7+XL208, and XL302+XL208. Procedures are similar to the circuit cross-talk experiment. Colonies were grown overnight in 4 ml LB with appropriate antibiotics and diluted 100 times on the next day, aliquoted in a 96-well plate. Inducer OC6 was added to wells (final max concentration 100 μM) and serial diluted three times. The 96-well plates were then incubated in a microplate (Lumitron) shaker for 210 min at 500 rpm.

*Pattern recognition experiments (Figs. 3c and 5c)*. For 4-bit patterns, relevant plasmids were XL140mut40+XL208, XL140mut8+XL208, XL140mut7+XL208, and XL340+XL291. For 3 × 3-bit pattern, relevant plasmids were XL140mut15+XL208, XL140mut7+XL208, XL302+XL208 and XL340+XL291. In both experiments, senders and receivers were grown overnight in 4 ml LB with the appropriate antibiotics. On the next day, senders were diluted 100 times in OC6 (final max concentration 33.3 μM) and aTc (final concentration 20 ng ml⁻¹) and incubated at 37 °C in a microplate shaker (Lumitron) at 500 rpm for 210 min. Next, the incubated solutions were diluted 100 times in LB containing antibiotics and incubated for another 30 min. The incubated solutions were mixed following the product of weights and patterns. The weight-pattern products are listed in Supplementary Fig. 20 for 3 × 3-bit patterns. Sender mixtures were transferred with receivers (50 times dilution from overnight culture) in a 1-to-19 volume ratio. The mixed senders and receivers were incubated in a microplate shaker (Lumitron) for 90 min at 500 rpm.

**Reporting summary**. Further information on research design is available in the Nature Research Reporting Summary linked to this article.

## Data availability

The FACS experiment data underlying Figs. 2b, d, e, 3c, 4c and 5c have been deposited to flowrepository with IDs FR-FCM-Z3CK, FR-FCM-Z3CW, FR-FCM-Z3D6, FR-FCM-Z3DG, FR-FCM-Z3MQ, and FR-FCM-Z3DD. These data are also available in GitHub with DOI [10.5281/zenodo.4682962] under "experiment_data" subdirectory. Plasmids with maps in Supplementary Fig. 23 are included in the same GitHub repository. All relevant data are available from the authors upon request. Source data are provided with this paper.

## Code availability

The algorithm code for weight optimization and experiment data analysis are available on GitHub with DOI [10.5281/zenodo.4682962].

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

## Acknowledgements

We thank Mark Epstein from Nati Korin's lab for helping illustrate the 3D model of the fluidic device as in Supplementary Fig. 7. Israel Science Foundation (ISF) [1558/17].

## Author contributions

R.D. and X.L. designed the study. R.D. and X.L. invented the ANN genetic circuits. X.L., L.R., and V.K. constructed plasmids. X.L. performed experiments and analyzed data. L.R. performed site-directed mutation on $P_{lux}$ and characterized the mutation library (Supplementary Fig. 3). V.K. performed site-directed mutation on $P_{cin}$ and characterized the mutation library (Supplementary Fig. 18). M.K. and N.K. helped with the design (Supplementary Fig. 7) and fabrication of fluidic devices. X.L. developed the algorithm and ran simulations. X.L. and R.D. discussed results and wrote the manuscript.

## Competing interests

The authors declare no competing interests.
