## [Peer Review File · Nature Communications]

Reviewers' Comments:

Reviewer #1:

Remarks to the Author:

In this manuscript, the authors propose a new architecture to program gene circuits for ANN-like computing in bacteria consortia for recognizing patterns; their genetic circuit mimics the structure of ANN using quorum sensing and decision making among a population of bacterial cells. The architecture is based on two types of bacteria, sender (for weights) and receiver (for summation-activation). Recognition of 3x3 patterns is experimentally tested.

Their simulations include:

- Recognition of more complicated patterns is simulated to verify the circuit performance on larger input.
 - The authors discussed methods for implementing the simple perceptron and optimizing the weights for pattern classification based on algorithm similar to gradient descent.
 - Potential issues regarding noise and crosstalk are clearly addressed. The authors also discussed several challenges such as the inherent noises of genetic circuits and how that affected the achieved results. Further, computer simulation was conducted.
- All these experimental and simulation results support their claims.

Although the architecture is novel and well demonstrated by experiments, there are still some aspects that need further explanation or improvement.

Edit Suggestions

- Redundant citation of [1] in first two sentences.

In the sentence lines 243-245 including "that couples design in silico and implementation in vivo", I feel it is more conventional to put the in silico and in vivo expressions in front. "in silico design and in vivo implementation".

- The inputs "0" and "1" are encoded by no OC6 and high OC6, respectively. Does it mean that this architecture cannot handle input "0" by a low (but nonzero) concentration of OC6? Is it possible to implement some kind of threshold such that a low (but nonzero) concentration of OC6 can still be treated as "0"?

- The weights can be tuned by mutating the first four base pairs in Plux. To my understanding, it means that the weight value is in a discrete manner. How will this property influence the computational power of this architecture? This issue should be explained or addressed at least in theory.

- According to Figure 3 (a), the bacteria for different pattern bits are operated separately. This will dramatically increase the operation complexity, especially for large-scale pattern recognition and potential future multilayer networks. Is it possible to do a one-pot experiment?

- Following comment 3, if it is not possible to put all parts together in one pot, the main issue is that all input bits are encoded by the same molecule (OC6). If we put all parts together, it will be impossible to distinguish input patterns with the same number of "1"s. It means that this architecture relies on separate reaction solutions for different pattern bits to differentiate input bits (that are all encoded by the same signal OC6). In the ideal case, input bits should be encoded by different chemical species that form a real "pattern". Currently, the networks essentially only catch one signal (OC6). The input pattern is not really encoded by concentrations of input species, but by separate handling of different pattern bits.

- Consider citing with [28] other works on DNA computation of cells, eg "Song, et al Programming DNA-based biomolecular reaction networks on cancer cell membranes. JACS, 2019. " .

For accurate classification, there appears to be a very hefty task of manually finding weights. The authors have supplied their results for such cases where they have already done the work, and I do not believe this detracts from the overall result. If there is still space in the manuscript, some thoughts on this topic as future work would be great.

Overall, the manuscript is interesting and may find potential applications in microbiome and biomedical engineering. But the edit requests need to be done before acceptance.

Reviewer #2:

Remarks to the Author:

This is a nice manuscript and one of the very few attempts to engineer machine learning devices in synthetic biology, for that reason the manuscript deserves publication provided the authors can address the following points and questions.

1) The context and motivation for the work are well introduced and literature properly surveyed. Fig1 is simple to follow and capture the essential of the circuitry involved, albeit some of the circuits becomes more complex in the subsequent Figures. For instance, how does relate the receiver circuit of Fig1.c to the open/PF circuits of Fig2.d?

2) Looking at Page 3 and Fig2.a, one notices there is no positive feedback for LuxR (under *placO* promoter instead of *plux*). With a perceptron one would like the outcome of the senders to be a weighted sum, where each sender, i , brings its own linear contribution $w_i \cdot x_i$ to the sum. Why then the sender circuit haven't been engineered to output a linear response (using PF regulation)? This particularly relevant as the technology to do that is well established and was first proposed by one of the authors of the current manuscript (ref. 9).

3) On page 3, the *placO* circuit is motivated by the following sentences "However, when mixing senders exposed to different inputs, i.e., with or without OC6, a cross-talk can occur between these senders . Specifically, senders that are initially inactivated ('0') may be affected by the residual OC6 from those that are activated ('1')". Following the exact same reasoning, senders that are initially activated ('1') may be affected by the presence of aTc. Could the authors comment on that? In particular and even though this is not mentioned on page 3, it looks like cross-talk is avoided because senders are incubated with OC6 or aTC independently (Fig3.b), they are then diluted 100 times before being mixed together and with the receiver with a 1 to 19 ratio. At the end the "cross-talk" circuit mentioned on page 3 looks to me as a device to just reduce the leakiness of *plux*. The authors should provide more explanations to motivate their "cross-talk" terminology.

4) On page 4, and Fig2.c-2.d, the way the sharpness of the sigmoid activation function is tuned is well introduced and explained, yet a link to the appropriate supplementary materials should be made here especially regarding Table 1. It would also be appropriate to summarize in the main text what are the main features of the model.

5) Regarding Fig2.e, the way weights are being increased (through mutation of the first four base pairs of *Plux*) is straightforward. However, I have difficulties to understand how the repressor works. I do not have issues with the *Plux* repressor system, nor with the plot that is indeed showing decrease in fluorescence level. However, as the repressor curve was plotted for an isolated sender, I am wondering what is happening when the senders are mixed together to ultimately perform a weighted sum (with one or several negative weights). A negative weight must decrease the amount of OHC14 produced by all senders and I am not seeing this happening here. Could the authors show evidences that overall response (when using multiple senders) is indeed decreased when using negative weights?

6) I have a question related to Fig3.a and the text on page 5 "There are in total twenty four 4-bit binary patterns, a subset of which can be well classified by the weight vector. " While it should be 16 and not 24, my question is why only 12 patterns have been tested? It would be appropriate to show the results for all and provide some explanations on why some could not be classified.

7) Moving onto Fig3.b, the experimental workflow introduces many numbers (time, dilution, volume). None of these are justified except perhaps the for 33um concentration chosen for OC6. What is the rationale for choosing the numbers that are reported? Were other tests carried out? Was a DoE type of approach used? This is particularly relevant for the various times indicated, for instance during the final incubation (90 min) could cross talk occur (see point 3 above)? Were other shorter or longer times tested and if so could the results be reported as well? It is also difficult to understand how negative weights can be implemented with such a workflow.

8) Fig4 and the corresponding text on page 6 are self-explanatory. However, I am a bit confused between the two-step and the one-step methods. The main differences between the 2 methods should be clearly stated in the main text and why they are used (or not) in specific cases. For instance, it appears that two-step method is used for classification of 3x3-bit patterns and one-step for 5x5, 7x7 and 9x9 bits, why?

9) Regarding Fig5 it would be wise to provide in the caption the pattern's numbers, this would help to understand Fig5.c, where one can guess the 4 first elements are from category z and the others from categories v and n. Also, the authors should provide explanation on why only 10 patterns are different among the 30 provided. I am not asking here to explain why some w.p products can be identical (this is well justified in the supplementary) but what are the limitations of the method that makes it so, and how it could be improved. Along these lines why mut8 and mut40 were not used for the 3x3 classification?

10) Finally, in the discussion, the authors should further discuss the advantages and differences of their technique in particular related to ref. 34, which like in the present manuscript is also engineering a perceptron capable of classifying input bit patterns. Also, the limitations of the technique need to be further listed and discussed, is there a maximum number of recognizable patterns for a nxn bits array? How can the limited number of patterns that are recognized be worked around? Can a multilayer perceptron be envisioned, how? Finally, what could be the practical applications of the proposed system and what are the hurdles to implement it.

Reviewer #3:

Remarks to the Author:

Authors present a proof-of-principle implementation of a simple single-layer artificial neural network (ANN) using multiple bacterial colonies to create a "perceptron". As input units, they used several colonies of bacteria endowed with OC6-inducible promoters driving expression of OC14 synthase ("sender" circuit). As input weights, they used strengths of the inducible promoters in different colonies. Each colony was cultured in a separate well, with different concentrations of OC6 added to each well to simulate the input binary pattern. After a defined period of time, each colony would produce a different amount of OC14 as determined by the amount of input OC6 and the strengths of promoters in each colony. Subsequent mixing of all colonies and their media played a role of summation of the weighted inputs. This mixture was then added to a separate well with bacteria endowed with the "receiver" circuit that has OC14-inducible promoter activating expression of a fluorescent protein that serves as readout. To increase the contrast of decision-making authors added a positive feedback loop to the circuit to increase the sharpness of the induction curve of the receiver circuit.

Their experiments showed that with appropriately chosen weights this pipeline can identify simple

4-bit or 9-bit (3x3) patterns reasonably well. This is a nice and convincing demonstration, and the results appear to be valid. However, I think the success of this demonstration is hardly surprising. Indeed, there is little doubt that the protocol described in the paper, if all the steps are performed accurately, would yield a functional perceptron-like classifier. Yet, a number of operations in this multi-step protocol needed to be performed manually: feeding the input signals (bits) into the separate wells with the "sender" colonies, deciding which input bits are "zero" and adding aTc to the corresponding wells. mixing cultures from sender wells and adding the mixture into the well with the "receiver" colony, and finally, collecting the receiver cells and running them through FACS. Conceptually, it is not that much different (albeit far more labor-intensive) from first measuring the input concentrations, either directly, or using a single reporter circuit (e.g. OC6-inducible GFP gene) with a tabulated induction curve, and then use a standard in silico perceptron to classify the pattern. Further, learning algorithm weights is implemented entirely in computer, and thus again is not much different from the standard ANN implementation. It would be far more interesting to devise a way for a bacterial consortium to learn and classify a pattern without major human intervention at the intermediate steps.

My main technical concern is about the use of aTc in "zero-bit" wells. As authors explain, its purpose is to avoid cross-talk between different sender colonies via residual OC6 after they are mixed together in the receiver well, because aTc would repress gene expression in the colonies which were exposed to it. Aside from the issue that it significantly diminishes the generality of the proposed classification scheme (we have to know a priori where there are "zeros" to add aTc there), I don't understand how this addresses the issue of cross-talk. Indeed, by the same logic, residual aTc would suppress gene expression not only in the well in which it was added, but in other senders as well, after they are mixed together. Furthermore, it seems that there is a much simpler way to eliminate this kind of cross-talk, namely, separate receiver cells from their media after cultivation in the presence of OC6, and only use media from different wells, not cells.

Overall, I think the paper lacks sufficient conceptual novelty that could make it suitable for Nature Communications. After an appropriate revision, it can probably be published in a more specialized journal, such as ACS Synthetic Biology.

Synthetic neural-like computing in microbial consortia for pattern recognition

Response to Reviewers Letter

We are grateful for the constructive feedback from all reviewers. Our responses to reviewers' comments are described in detail as below. *Reviewer comments are italicized.* **Our answers are in bold.**

REVIEWER COMMENTS

Reviewer #1 (Remarks to the Author)

In this manuscript, the authors propose a new architecture to program gene circuits for ANN-like computing in bacteria consortia for recognizing patterns; their genetic circuit mimics the structure of ANN using quorum sensing and decision making among a population of bacterial cells. The architecture is based on two types of bacteria, sender (for weights) and receiver (for summation-activation). Recognition of 3x3 patterns is experimentally tested.

Their simulations include:

- Recognition of more complicated patterns is simulated to verify the circuit performance on larger input.*
- The authors discussed methods for implementing the simple perceptron and optimizing the weights for pattern classification based on algorithm similar to gradient descent.*
- Potential issues regarding noise and crosstalk are clearly addressed. The authors also discussed several challenges such as the inherent noises of genetic circuits and how that affected the achieved results. Further, computer simulation was conducted.*

All these experimental and simulation results support their claims.

We thank the reviewer for the valuable feedback, which helped us further clarify and emphasize the contributions of our work. We provided comprehensive responses to the points raised by the reviewer as below.

Although the architecture is novel and well demonstrated by experiments, there are still some aspects that need further explanation or improvement.

Edit Suggestions

- Redundant citation of [1] in first two sentences.

We agree with the reviewer and have edited the citation.

In the sentence lines 243-245 including “that couples design in silico and implementation in vivo”, I feel it is more conventional to put the in silico and in vivo expressions in front. “in silico design and in vivo implementation”.

We agree with the reviewer and edited these lines. They are now on page 9, lines 287-288.

- The inputs “0” and “1” are encoded by no OC6 and high OC6, respectively. Does it mean that this architecture cannot handle input “0” by a low (but nonzero) concentration of OC6? Is it possible to implement some kind of threshold such that a low (but nonzero) concentration of OC6 can still be treated as “0”?

We appreciate the reviewer for this concern and discuss it in several cases. Yes, it is possible to implement a threshold level for input bits using a low analog input with nonzero inducer concentration. As shown from the transfer functions of sender circuits (Fig 2e), we can select an OC6 concentration approximately at $0.05 \mu M$ as the threshold level that imposes minimal effects on activating senders such as P_{lux} mut7, mut15 and mut8. For senders with high maximal activities such as mut40, it will be more challenging to implement a nonzero input, due to the higher basal levels.

Furthermore, in our protocol, there is another complication in implementing nonzero inputs in experiments. This complication arises from the fact that we used aTc for “0” input states to turn off the activity of constitutive promoter P_{lacO} on the low-copy-number plasmid, in order to prevent cross-talks when senders exposed to different levels of OC6 are mixed. To implement the nonzero input, when aTc is present, even a high concentration of OC6 would not activate P_{lux} promoters. A low threshold input is thus not applicable to this scenario.

Nevertheless, as a simple demonstration of low “0” inputs, we experimentally examined the circuit performance to a 2-bit pattern set (Fig S1). The inputs were two different chemical inducers, Arabinose and OC6. In this case, circuits for the two inducers are independent. There is no cross-talk and no requirement for aTc dependent circuit as in Fig 2a to control cross-talks. A nonzero level of inducers at $1 \mu M$ was used for “0” states, and $100 \mu M$ for “1” states. As shown in Fig S1, orange bars represent receiver output when no inducer is provided at “0” state and blue bars are for nonzero inducer at “0” state. Compared to the former case, receiver output corresponding to the latter exhibit higher levels at “00” and “01” patterns, but are still capable of separating patterns “00” and “01” versus patterns “10” and “11”.

To demonstrate a threshold input without using aTc, we simulated the circuit classifi-

cation results using simulations by varying the inducer concentrations for “0” bits from 0.02 to 0.1 μM and simulated the classification results (Fig S2). As expected, up to 0.05 μM , the classification results stay unaffected. The performance becomes compromised, as the inducer concentration for “0” bits increases. We included the above discussion in Supplementary Notes section Analog (non-zero) inputs.

- *The weights can be tuned by mutating the first four base pairs in P_{lux} . To my understanding, it means that the weight value is in a discrete manner. How will this property influence the computational power of this architecture? This issue should be explained or addressed at least in theory.*

We agree with the reviewer’s comment. Our implementation by mutating the first four base pairs of P_{lux} promoter can lead to at most 256 ($=4^4$) distinct mutations, which potentially provide a range of promoter strengths. We also have measured more than four mutated P_{lux} promoters, which leads to almost continuous weight values (Fig S3). We added this notion in the main text (results) page 5 line 132.

Despite so, the weights are discrete. In the following narrative, we show that the perceptron can be trained using discrete weights. Conventional gradient descent algorithm assumes continuous weight variables. To handle discrete weights, we introduced a hidden variable vector \vec{h} , which can be continuously updated [1]. Then the weight vector \vec{w} is adjusted depending on the values of \vec{h} (Fig S4).

We showed that classification is possible with a few discrete weight levels. Using simulations, we also examined a minimal size of weight set, showing that only two levels are potentially sufficient for classification (Fig S5). However, one caveat of a small weight set is that the classification performance is sensitive to noise in genetic circuits. A mutated P_{lux} promoter may exhibit varying strength from trial to trial, depending on variations in cell growth. To demonstrate the effects of such noise on classifications, using simulations we further showed that increasing available weight levels (from 2 levels to 5 levels) allows the perceptron to tolerate noise better (Fig S6c and d). The simulations indicate that a level of redundancy in weights is beneficial for the genetic perceptron to overcome the noise. We included this idea on discrete weights in the main text (results) page 7, lines 223 to 227. and Supplementary Notes section Algorithm development.

We are also glad to notice that the idea of discrete weights has been used in other architectures. In machine learning, traditional algorithms such as gradient descent treat weights as continuous variables. However, recent studies in this field investigated new frameworks such as binarized neural networks (i.e., binary weights and activation) [2], to reduce the storage requirement for high precision parameters and improve the operation speed.

- *According to Figure 3 (a), the bacteria for different pattern bits are operated separately. This*

will dramatically increase the operation complexity, especially for large-scale pattern recognition and potential future multilayer networks. Is it possible to do a one-pot experiment?

We appreciate the reviewer for the insight. It is possible to interpret each bit or pixel as a distinct chemical inducer, and perform a “one-pot” experiment using circuits constructed to sense these chemical inputs (Fig S1). As discussed in a previous comment, we constructed a circuit to sense Arabinose, using the same topology as the circuit for OC6. Two types of sender circuits were mixed with or without corresponding inducers, and incubated with receivers for three hours in the same reaction compartment. Afterwards, we examined the activities of receivers, showing that receivers were activated to distinct levels, in response to combinations of sender patterns.

Alternatively, since the patterns in Fig 3a contain spatial information, reactions from different pixels or bits operate with spatial separation. One option to scale up the architecture while keeping spatially separated reactions is to use devices such as microfluidics. In this case, individual reactions can be compartmentalized and the entire process can be completed without heavy manual operations. As a demonstration, we implemented a simple fluidics device for 2-bit patterns (Fig S7-9).

We elaborated this point in the main text (discussion) page 9, lines 289 to 301, and included the discussion in Supplementary Notes section Alternative implementations.

- Following comment 3, if it is not possible to put all parts together in one pot, the main issue is that all input bits are encoded by the same molecule (OC6). If we put all parts together, it will be impossible to distinguish input patterns with the same number of “1”s. It means that this architecture relies on separate reaction solutions for different pattern bits to differentiate input bits (that are all encoded by the same signal OC6). In the ideal case, input bits should be encoded by different chemical species that form a real “pattern”. Currently, the networks essentially only catch one signal (OC6). The input pattern is not really encoded by concentrations of input species, but by separate handling of different pattern bits.

We appreciate the reviewer for pointing out the spatial assumption in the architecture. However, as explained in the previous question, the architecture can be adapted to handle patterns of multiple-molecule species by interpreting separate pixels as distinct chemical species. As discussed previously, we demonstrated the feasibility of using a set of 2-bit patterns (Fig S1).

To recognize more types of chemical inputs, essentially we need to include multiple different parts into our architecture, such as promoter sequences or signaling components. Meyer et al. [3] has engineered an *E. coli* strain capable of optimally sensing 12 types of small molecules. Studies as such provide useful resources for us to expand the architecture to handle more complex patterns.

- Consider citing with [28] other works on DNA computation of cells, eg “Song, et al Programming

DNA-based biomolecular reaction networks on cancer cell membranes. JACS, 2019. "

Thank you for bringing up the relevant reference. We have included it in the main text (discussion) on page 10 line 317.

For accurate classification, there appears to be a very hefty task of manually finding weights. The authors have supplied their results for such cases where they have already done the work, and I do not believe this detracts from the overall result. If there is still space in the manuscript, some thoughts on this topic as future work would be great.

We suppose the reviewer concerns that weights are obtained through manually introduced mutations. In this regard, our present architecture does not require weights of fine resolutions. Two or three distinct weight levels are sufficient to perform classification (Fig S5 and S6). Yet we are aware of the *in vitro* mutation step in our method. In the future, it is possible to implement adaptive weights update *in vivo*, which may rely on *in vivo* base editing [4].

On the other hand, the reviewer might concern the manual procedures involved in the protocol. In this regard, we agree that devices such as micro-fluidics can be employed in the future to classify large-scale patterns. As mentioned previously, we demonstrated the idea of using a simple fluidics device containing three wells (Fig S7). The classification task was performed within two hours, requiring manual steps only at the very beginning.

Overall, the manuscript is interesting and may find potential applications in microbiome and biomedical engineering. But the edit requests need to be done before acceptance.

We thank the reviewer again the comments.

Reviewer #2 (Remarks to the Author)

This is a nice manuscript and one of the very few attempts to engineer machine learning devices in synthetic biology, for that reason the manuscript deserves publication provided the authors can address the following points and questions.

We thank the reviewer for the valuable comments, which are helpful for us to further clarify our work. We provided the responses to the points raised by the reviewer.

1) The context and motivation for the work are well introduced and literature properly surveyed. Fig1 is simple to follow and capture the essential of the circuitry involved, albeit some of the circuits becomes more complex in the subsequent Figures. For instance, how does relate the receiver circuit of Fig1.c to the open/PF circuits of Fig2.d?

Thank you for the suggestion. We have updated Fig 2d to show the open-loop (OL) and positive feedback (PF) circuits separately. The basic receiver circuit in Fig 1c is an OL construct. The PF circuit in Fig 2d contains one additional part P_{cin} -CinR on a low-copy-number plasmid to increase the steepness of transfer function. We also edited the caption of Fig 2 to clarify how Fig 2d relates to Fig 1c.

2) Looking at Page 3 and Fig2.a, one notices there is no positive feedback for LuxR (under $placO$ promoter instead of $plux$). With a perceptron one would like the outcome of the senders to be a weighted sum, where each sender, i , brings its own linear contribution $w_i \cdot x_i$ to the sum. Why then the sender circuit haven't been engineered to output a linear response (using PF regulation)? This particularly relevant as the technology to do that is well established and was first proposed by one of the authors of the current manuscript (ref. 9).

We thank the reviewer for the insight. In our case, the mutated P_{lux} promoters exhibit transfer functions that closely resemble linear functions (Fig 2e). As shown in Table 1, Hill coefficients for this mutated P_{lux} are far below 1. Moreover, all transfer functions can be approximated using the same set of parameters (K_d , n , and β_0), with the maximal activity β_m being the defining variable. This is convenient for us to model the network using maximum activity β_m as a weight variable.

As suggested by the reviewer, we implemented sender circuits with PF regulation (Fig S10a) by replacing the P_{lacO} promoter with a mutated P_{lux} that matches with the promoter driving the *cinI* and *mCherry* genes. The transfer functions of PF circuits were fitted using Hill equations, and the fitting parameters vary depending on P_{lux} mutations (Table S3). With these specific mutations, the circuit behaviors differ from the linearity shown in ref 9 because an additional shunting part, such as a high-copy-number plasmid (HCP) with P_{lux} promoter, is necessary to sequester extra transcription factors. In ref. 9, the signal part was located on a HCP or MCP for shunting to prevent the signal from saturating. Thus, there need to be additional regulatory components, if to use PF regulation to

achieve linearity.

Nevertheless, the PF circuits exhibit higher activation thresholds than OL circuits. This leads to a desirable property that cross-talks between senders can be managed by merely diluting the senders 100 times, and the residual inducer concentration can be reduced below the threshold. Thus, aTc is not required to minimize cross-talks. We verified this property using an assay of 4-bit patterns as shown in Fig S10b. Furthermore, despite their nonlinear transfer functions, the PF sender circuits can classify 4-bit patterns (Fig S11). In this case, due to the steep transfer functions, PF circuits are sensitive to inducer concentrations. Thus, a proper inducer concentration for “1” states needs to be chosen carefully to keep senders in the correct activation level.

We elaborated this point in Supplementary Notes section Cross-talk discussion and Alternative implementations.

*3) On page 3, the *placO* circuit is motivated by the following sentences “However, when mixing senders exposed to different inputs, i.e., with or without OC6, a cross-talk can occur between these senders. Specifically, senders that are initially inactivated ('0') may be affected by the residual OC6 from those that are activated ('1')”. Following the exact same reasoning, senders that are initially activated ('1') may be affected by the presence of aTc. Could the authors comment on that?*

In our study, both residual OC6 and aTc can affect senders. In the main text, we are primarily concerned about the cross-talk from OC6 to “0” state senders with P_{lux} activators (positive weight). In the same way, residual OC6 can also repress “0” state senders with P_{lux} repressors (negative weight).

We agree with the reviewer that aTc can also affect senders. We illustrated the possible effects in Fig S12a for an explanation. Senders with P_{lux} activators (the orange line) and P_{lux} repressors (the blue line) are incubated with OC6 first and then diluted in exposure to aTc. Hypothetically, for positive weights senders at “1” state, exposure to aTc tends to reduce promoter activity, due to a decreasing activity of P_{lacO} . This process would result in a slowed production of OHC14. However, this effect would not significantly decrease the amount of OHC14, because lactones are highly stable, estimated based on a slow degradation rate of OC6 [5].

For senders containing P_{lux} repressors, turning off P_{lacO} by aTc tends to reactivate repressor promoters. Nonetheless, the residual aTc at 0.2 ng ml^{-1} does not reactivate P_{lux} repressors until approximately two hours. These temporal dynamics are shown in Fig S12b by time-course data collected from a plate reader. In the assay, we incubated P_{lux} repressor senders in OC6 ($100 \mu\text{M}$) for four hours. Senders were subsequently diluted 100 times and incubated in aTc with varying concentrations. The OD600 and mCherry levels were monitored across time at every 15 min. We noticed an increase in mCherry level around 150 min. In our protocol, the sender mix was incubated with

receivers for 90 min, which is within the safe period of two hours.

To experimentally demonstrate that aTc will not reactivate P_{lux} repressor senders, we carried out an assay by mixing P_{lux} activator senders incubated with aTc (senders at “0” state) and P_{lux} repressor senders incubated with OC6 (senders at “1”). In the three 4-bit patterns (Fig S12c), all bits are expected to generate low OHC14. Thus, unless P_{lux} repressors become reactivated, all patterns should result in low signal output in receivers. After incubation, we examined whether receivers produce EYFP due to the possible reactivation of P_{lux} repressor senders. As expected, no activation of receivers was observed (Fig S12c).

We included the above analysis in Supplementary Notes section Cross-talk discussion for clarification.

In particular and even though this is not mentioned on page 3, it looks like cross-talk is avoided because senders are incubated with OC6 or aTC independently (Fig3.b), they are then diluted 100 times before being mixed together and with the receiver with a 1 to 19 ratio. At the end the “cross-talk” circuit mentioned on page 3 looks to me as a device to just reduce the leakiness of lux . The authors should provide more explanations to motivate their “cross-talk” terminology.

The cross-talk among senders indeed exists. We performed an assay for demonstration. We designed three 4-bit patterns (in Fig S13) using weights of mut40 and mut8. The “1” states inputs are given to mut8, which is a weak mutated P_{lux} promoter and generates a small amount of OHC14. All bits of the three patterns should correspond to low overall OHC14. Thus, all three patterns should result in low receiver activities. We examined the corresponding receiver output for the three patterns when senders were pre-incubated with and without aTc (20 ng ml^{-1}). Without aTc, sender mix lead to higher activation levels in receivers than with aTc. We included this notion in Supplementary Notes section Cross-talk discussion.

4) On page 4, and Fig2.c-2.d, the way the sharpness of the sigmoid activation function is tuned is well introduced and explained, yet a link to the appropriate supplementary materials should be made here especially regarding Table 1. It would also be appropriate to summarize in the main text what are the main features of the model.

Thank you for pointing this out. We added short descriptions in the main text to explain how transfer functions were fitted using Hill equations. On page 4 line 120, we added the following content. “We fitted the transfer functions of the two circuits using a Hill equation, in the form $f(x) = \beta_m \frac{(x/K_d)^n}{1+(x/K_d)^n} + \beta_m \beta_0$. Here x represents the inducer amount, K_d is the dissociation constant between the inducer-TF complex and the promoter, β_m is the max activity of the promoter, $\beta_m \beta_0$ describes the basal activity of the promoter, and n is the Hill coefficient, representing the cooperativity involved in TF-inducer binding and TF-promoter binding.”

On page 5 line 142, we added “The P_{lux} activators were fitted using the same equation as described previously for receiver circuits. Notice that K_d , n and β_0 are similar for different mutations, leaving β_m as the defining variable for the transfer functions. Therefore, we factor out β_m and consider it as the weight variable. The P_{lux} repressor was fitted using an equation in the form $f(x) = \alpha_m \frac{1}{1+(x/K_d)^n} + \alpha_0 \alpha_m = \hat{\alpha}_m (\frac{(x/K_d)^n}{1+(x/K_d)^n} - \alpha_0 - 1)$, where K_d and n are the same as previously described. Similar to β_m and $\beta_m \beta_0$, α_m and $\alpha_m \alpha_0$ describe the max activity and minimal activity of the P_{lux} repressor promoter. $\hat{\alpha}_m$ is the weight for the repressor promoter and has a negative value, consistent with the negative slope of the transfer function.”

Using sensitivity analysis, we further demonstrated that the transfer function fittings of P_{lux} activator promoters are most sensitive to the parameter β_m . Thus, we can simplify the corresponding transfer functions by setting parameters other than β_m as constant and single out β_m as the weight variable. We included further discussion regarding sensitivity analysis (Fig S14) in Supplementary Notes section Transfer function fitting.

5) Regarding Fig2.e, the way weights are being increased (through mutation of the first four base pairs of P_{lux}) is straightforward. However, I have difficulties to understand how the repressor works. I do not have issues with the P_{lux} repressor system, nor with the plot that is indeed showing decrease in fluorescence level. However, as the repressor curve was plotted for an isolated sender, I am wondering what is happening when the senders are mixed together to ultimately perform a weighted sum (with one or several negative weights). A negative weight must decrease the amount of OHC14 produced by all senders and I am not seeing this happening here. Could the authors show evidences that overall response (when using multiple senders) is indeed decreased when using negative weights?

In general, we considered two possible approaches for implementing negative weights. The first approach is using a repressor promoter, as we implemented in the study. In the absence of inducers, the promoter activity is high. Thus, a summation of the negative weight and positive weight promoters results in an amount of OHC14 in an approximate form $w_1 x_1 + (const - w_2 x_2)$, where w_1 and w_2 are positive and negative weights, respectively. This implementation leads to an approximate linear summation of OHC14.

A second approach involves degradation proteins. OHC14 can be inactivated by an enzyme N-acyl homoserine lactonase encoded in the gene *aiiA*. Thus, a possible solution to decrease the amount of OHC14 is to express the gene *aiiA*. However, this approach has two flaws. First, the enzyme is not specific to OHC14 but can degrade a range of lactones, including the input inducer OC6. Second, unlike lactones, the enzyme encoded in gene *aiiA* cannot diffuse to the outside of cells. This adds to the complexities of implementation to achieve linear summation of OHC14.

We adopt the first approach for implementation. In this approach, positive weight refers to the maximum promoter strength variable β_m in a fitted transfer function of an

activator promoter, which has a positive value and describes the upward slope of the transfer function. For a repressor promoter, the corresponding weight variable is $\hat{\alpha}_m$, which has a negative value. Intuitively, a negative weight describes the downward slope of the repressor promoter transfer function. A repressor promoter shows high promoter activity in the absence of the inducer and low promoter activity in the presence of the inducer. Thus, when mixing senders of positive and negative weights, whether or not a negative weight decreases the overall amount of OHC14 depends on the input patterns.

To demonstrate experimentally, we examined a set of 2-bit patterns using a negative weight and a mutated positive weight. The two senders were incubated separately with or without inducers and mixed according to four pattern combinations (Fig S15). The mCherry levels of sender mix solutions were measured using flow cytometry. Among the four patterns, pattern "10" (OC6 for positive weight sender only) results in the highest overall mCherry, whereas pattern "01" (OC6 for negative weight sender only) results in the lowest overall mCherry. Thus, the effect of negative weights depends on inducer patterns. Their presence can add to the overall promoter activities when they are not exposed to OC6.

We included the above discussion in Supplementary section Negative weight.

6) *I have a question related to Fig3.a and the text on page 5 "There are in total twenty four 4-bit binary patterns, a subset of which can be well classified by the weight vector. " While it should be 16 and not 24, my question is why only 12 patterns have been tested? It would be appropriate to show the results for all and provide some explanations on why some could not be classified.*

Regarding the complete 4-bit patterns, we thank the reviewer for pointing out the typo. We corrected it in the main text and examined all 16 patterns (Fig S16). The four additional patterns are placed in the middle of the graph (grey bars, patterns '0011', '0110', '1001', and '1100').

As shown in Fig 3a, we used weight values [450, 3500, 900, 3500] and calculated summation of mCherry values for the 16 patterns. We picked a lower bound 4000 in summed mCherry value and a higher bound 4500 to separate patterns into distinct categories. In Fig 3a, the corresponding summed mCherry levels are listed beside each pattern. The overall mCherry levels for the additional four patterns equal 4400, 4400, 3950, and 3950 respectively, which overlap with the region between two bound values and thus are not easy to classify. This is an issue inherently present in ANNs. Therefore, 12 patterns were presented in Fig 3. Nevertheless, as in Fig S16, if arbitrarily choosing an intermediate value as the single boundary, our experiment results show that the data can be classified into two states.

We included the above analysis in Supplementary Notes section Explanation on classifying 4-bit patterns.

7) Moving onto Fig3.b, the experimental workflow introduces many numbers (time, dilution, volume). None of these are justified except perhaps the for 33um concentration chosen for OC6. What is the rationale for choosing the numbers that are reported? Were other tests carried out? Was a DoE type of approach used? This is particularly relevant for the various times indicated, for instance during the final incubation (90 min) could cross talk occur (see point 3 above)? Were other shorter or longer times tested and if so could the results be reported as well? It is also difficult to understand how negative weights can be implemented with such a workflow.

Regarding the workflow, three duration parameters are particularly relevant. First is the sender incubation period. We expect it to be long enough for cell growth so that activator senders can reach proper activities and levels of repressor senders can be sufficiently reduced. Meanwhile, the incubation should not be excessively long such that over amount of cells may result in inducer being effectively diluted and losing effects, especially for inducers of TF repressors. We measured the cell growth in OD600 across time using a plate reader (Fig S17). Based on the slopes of growth curves, the fast growth period ends between 200 to 300 min.

After diluting incubated senders 100 times, we allow senders to incubate for another 30 min, which is approximately one cell cycle. This step serves as a slight adjustment to OD after dilution so that OD levels can reach approximately 0.1, within the measurable range of the plate reader. All sender solutions should be of roughly equal OD levels.

The third duration parameter is the incubation time between sender mix and receivers. In comment 3, we have justified that a safe period of two hours has been considered to prevent inactivated P_{lux} repressor promoters become reactivated. Our protocol using 90 min is within a safe period. To ensure sufficient cell densities for flow cytometry measurement, receivers were initially diluted 50 times from overnight culture. The volume ratio of sender to the receiver was selected after trials. These numbers can be adjusted depending on cell growth conditions.

We included the above discussion in Supplementary Notes section More on experimental workflow.

8) Fig4 and the corresponding text on page 6 are self-explanatory. However, I am a bit confused between the two-step and the one-step methods. The main differences between the 2 methods should be clearly stated in the main text and why they are used (or not) in specific cases. For instance, it appears that two-step method is used for classification of 3×3 -bit patterns and one-step for 5×5 , 7×7 and 9×9 bits, why?

For 3×3 patterns, the gradient-based method alone does not necessarily converge to the optimal solution, therefore we used step2 that involves a direct search to further optimize weights. Regarding when the direct search is adopted, we added in the main text on page 8 lines 239 to 241 that "The three vectors can be obtained by first following

the gradient-based method and then a direct search, since the gradient-based approach alone does not necessarily converge to the optimal solution". In case of large patterns, gradient-based method alone can lead to very good classification results, either assuming weights take discrete levels (Fig 6) or arbitrary continuous values (Fig S21). This observation is partly related to the sparseness of patterns, that is, the probability of an arbitrary pixel in a pattern is "0" state. Here we use the proportion of "0" state bits relative to the total pattern size to capture the concept. For 3×3 "z", "v" and "n" patterns, the percentage of "0" states is $\frac{3}{9}$ ($= 0.33$); for 5×5 , 7×7 and 9×9 patterns, the percentages are $\frac{16}{25}$ ($= 0.64$), $\frac{36}{49}$ (≈ 0.73) and $\frac{64}{81}$ (≈ 0.79), respectively. Theoretical works indicate that capacity of neural network increases for sparser patterns [Bi_2020]. We added this notion to the main text (results) on page 8, lines 262 to 267.

9) Regarding Fig5 it would be wise to provide in the caption the pattern's numbers, this would help to understand Fig5.c, where one can guess the 4 first elements are from category z and the others from categories v and n. Also, the authors should provide explanation on why only 10 patterns are different among the 30 provided. I am not asking here to explain why some $w \cdot p$ products can be identical (this is well justified in the supplementary) but what are the limitations of the method that makes it so, and how it could be improved. Along these lines why mut8 and mut40 were not used for the 3x3 classification?

Thank you for the suggestion in Fig 5. We have updated the Fig 5b and c to explain the repetitions involved in pattern classification.

Regarding that only 10 patterns among the 30 are distinct, there are two factors rendering such repetitions. First, three distinct weight values are used for classification. To classify 3×3 patterns in binary values, essentially a few weights are sufficient to provide good classifications (Fig S5). With only a few weight values, many $\vec{p}^T \vec{w}$ products essentially consist of repeating elements. Combining elements by weights allow us to reduce the size of patterns. Second, these elements can be combined because we directly mix senders during the experiment. In this case, how we mix the senders, that is, the order of elements in $\vec{p}^T \vec{w}$ products does not matter. We included the above explanation in Supplementary Notes section Repetitions in patterns.

In our design, one limitation is that noise in genetic circuits can aggregate during the summation of multiple inputs. Hypothetically, when reaching a certain number of inputs, the receivers cannot necessarily distinguish the on/off ratio in one input bit versus noise fluctuation. Thus, the maximum number of inputs depends on the dynamic range of receivers, and the noise fluctuation in inputs. A second limitation is related to the cascades of multiple network layers. In this case, the dynamic range as well as the sharpness of the transfer function for each layer requires optimization. Also, noise propagation across layers can limit the operation of networks [6, 7]. We expect to characterize these network properties in future studies. A third limitation is regarding the basal level of the system,

which is not strictly zero. This issue can be potentially solved by adjusting the ratio of senders to receivers or modifying the threshold of the receivers. We can adjust the threshold of receiver promoters by mutating the first three base-pairs of the promoter operator (Fig S18). We included the notion regarding noise in the main text (discussion) on page 10, lines 306 to 313.

Regarding how the weights were determined, using our algorithm with two steps, we can calculate a weight vector for \vec{w}_0 as [1813, 1813, -735, -735, 2872, -735, -735, 1813, 1813] (Table S4). To find out the corresponding positive weights, we check the β_m values in Table 1, from which we can tell a value closest to 1813 is mut7 (1906) and a value closest to 2871 is mut15 (2679). Thus, we do not need to use mut8 and mut40. We also implemented a new model to update only discrete weight levels (Fig S4). In this model, with a given list of weight values, the algorithm can return a combination of these weights that yields good classification.

10) Finally, in the discussion, the authors should further discuss the advantages and differences of their technique in particular related to ref. 34, which like in the present manuscript is also engineering a perceptron capable of classifying input bit patterns.

The study by Pandi et al. implemented a metabolic perceptron that can classify 4-bit inputs. In their work, a few aspects are different from our study.

- First, the summation is carried out in single cells or cell-free systems in Pandi et al. Transducers that take inputs and actuators that report output is located in the same compartment. In our study, input uptake locates in senders and summation is in receivers, which are two separate cell groups. From a design perspective, separate cell populations taking different tasks are functional modules and allow for scale-up to more complicated networks.
- Second, in Pandi et al., individual bits of patterns are represented as distinct transducers. In contrast, in our study, pattern bits are pixels of spatial patterns and as concentrations of inducer levels. Although a single type of inducer OC6 was used, our system can also be adapted to take different types of chemical inducers (Fig S1).
- Third, we implemented a positive feedback regulation in the receiver circuit to improve the sharpness of the transfer function. Yet no such regulation is employed in Pandi et al.. In particular, the transfer function of their system *in vitro* (Fig 3c Pandi et al.) seems sharper than *in vivo* (Fig 1c). This observation might be related to substrate saturation in cell-free systems. This mechanism is unavailable in living cells. Also, nonlinear decision-making based on such a mechanism can be challenging to scale up.

- Fourth, in Pandi et al., weights are represented as concentrations of DNA that encodes enzymes in the cell-free system. These weights are positive continuous numbers. In our study, however, weights are not constrained to positive values. We implemented weights by mutating base pairs in P_{lux} promoter. In particular, we implemented P_{lux} repressor promoters, which function as negative weights and are highly useful for classifying certain patterns (Fig S19).
- Most importantly, while Pandi et al. demonstrated their method using logic functions, we implemented complex pattern recognition based on ANN-like design. We also presented an algorithm to update weights. Our framework allows for sophisticated classification tasks and capability of scale-up.

Also, the limitations of the technique need to be further listed and discussed, is there a maximum number of recognizable patterns for a $n \times n$ bits array?

Mathematically, there is an upper limit of recognizable patterns for a $n \times n$ bits array. A perceptron can recognize only linearly separable patterns. A function counting theorem [8] states that for p random patterns and N synapses (or weights, here $N = n^2$ for a $n \times n$ bits array), the total number of arbitrary binary labels on these patterns that allow linear separability is $C(p, N) = 2 \sum_{i=0}^{N-1} \binom{p-1}{i}$. As N increases, $C(p, N)$ increases as well, but the ratio of this number to the total possible binary labels 2^p drops quickly after p becomes larger than $2N$. When p equals $2N$, half of the binary labels are linearly separable and can be realized using perceptrons. Thus for large N , the value 2 bits per synapse has been considered as the capacity of a perceptron with N synapses [8].

The storage capacity of correlated patterns, as in our case for patterns within the same class, is generally higher [9]. When patterns and weights are restricted to binary values, the capacity decreases and were estimated with a maximum of 0.83 bit per synapse [10]. The sparseness of patterns also affects capacity. In particular, capacity is improved for patterns with high sparseness [1].

Aside from the mathematical limit in capacity, the size of patterns that can be classified by our design is limited biophysically by noise in genetic circuits. As stated previously in comment 9, such noise can aggregate during the summation of multiple inputs. Hypothetically, when reaching a certain number of inputs, the receivers cannot necessarily distinguish the on/off ratio in one input bit versus noise fluctuation. Thus, the maximum number of inputs depends on the dynamic range of receivers and the noise fluctuation in inputs.

How can the limited number of patterns that are recognized be worked around? Can a multilayer perceptron be envisioned, how?

Since the limited number of patterns discussed above is inherent to perceptron, constructing multi-layer networks can overcome the limitation of linear separability and ex-

pand the repertoire of recognizable patterns. Increasing the network layers certainly adds the complexity of the architecture to classify more sophisticated patterns. This has been shown by Tamsir et al [11] that a second layer cascade allows the genetic circuit to compute XOR. However, multi-layer networks also require signaling molecules that are mutually orthogonal to reduce cross-talks between network layers. Recent work by Kylilis et al. [12] on a library of quorum sensing lactone devices would help to design and construct multi-layer networks. Moreover, the transfer function of each layer requires adjustment to optimize the dynamic range. Fine-tuning each layer can be realized by mutating relevant promoters and modifying the circuit topology via regulations. We included the above discussion in Supplementary Notes section Capacity of perceptron and scale-up to multi-layer networks.

Finally, what could be the practical applications of the proposed system and what are the hurdles to implement it.

The proposed system can be used to facilitate the design of living therapeutics. For example, recently Gurbatri et al. [13] engineered a probiotic bacteria system to control the release of immune checkpoint inhibitors for cancer therapy. The system adopts a genetic circuit based on quorum sensing to synchronize drug release. In the future, our framework can help design the drug release system to achieve more sophisticated control.

The proposed system can also be potentially used for engineering cellular decision-making systems based on inter-cellular communications. For example, in living systems, tissue cells can self-organize into different spatial patterns by sensing local concentrations of signaling molecules [14]. The decision-making involved in the process can be programmed based on our design to control morphogenesis.

In addition, our framework provides a prototype to implement more sophisticated computations based on collective activities from cell groups. Apart from perceptron, there are more complicated network structures that enable the representation of sensory information, which can be investigated to develop advanced functions in the future.

We have included the above discussion in the main text (discussion) on page 10 lines 339 to 348.

Reviewer #3 (Remarks to the Author)

Authors present a proof-of-principle implementation of a simple single-layer artificial neural network (ANN) using multiple bacterial colonies to create a “perceptron”. As input units, they used several colonies of bacteria endowed with OC6-inducible promoters driving expression of OC14 synthase (“sender” circuit). As input weights, they used strengths of the inducible promoters in different colonies. Each colony was cultured in a separate well, with different concentrations of OC6 added to each well to simulate the input binary pattern. After a defined period of time, each colony would produce a different amount of OC14 as determined by the amount of input OC6 and the strengths of promoters in each colony. Subsequent mixing of all colonies and their media played a role of summation of the weighted inputs. This mixture was then added to a separate well with bacteria endowed with the “receiver” circuit that has OC14-inducible promoter activating expression of a fluorescent protein that serves as readout. To increase the contrast of decision-making authors added a positive feedback loop to the circuit to increase the sharpness of the induction curve of the receiver circuit.

Their experiments showed that with appropriately chosen weights this pipeline can identify simple 4-bit or 9-bit (3x3) patterns reasonably well. This is a nice and convincing demonstration, and the results appear to be valid.

We thank the reviewer for the comments.

However, I think the success of this demonstration is hardly surprising. Indeed, there is little doubt that the protocol described in the paper, if all the steps are performed accurately, would yield a functional perceptron-like classifier.

We have made a substantial effort to optimize the protocol. Now our results are reproducible with some expected error margin, which is typical in biological experiments conducted in wet lab.

Yet, a number of operations in this multi-step protocol needed to be performed manually: feeding the input signals (bits) into the separate wells with the “sender” colonies, deciding which input bits are “zero” and adding aTc to the corresponding wells. mixing cultures from sender wells and adding the mixture into the well with the “receiver” colony, and finally, collecting the receiver cells and running them through FACS. Conceptually, it is not that much different (albeit far more labor-intensive) from first measuring the input concentrations, either directly, or using a single reporter circuit (e.g. OC6-inducible GFP gene) with a tabulated induction curve, and then use a standard in silico perceptron to classify the pattern.

We agree with the reviewer. In our revised manuscript, we added an implementation for cell communication based on micro-fluidics devices (Fig S7). The new implementation involves a few manual steps. We demonstrated using a three-well fluidics device, which can recognize patterns of 2-bit inputs (Fig S8-9). We expect to scale-up the fluidics

devices in the future to handle more complicated patterns.

Further, learning algorithm weights is implemented entirely in computer, and thus again is not much different from the standard ANN implementation. It would be far more interesting to devise a way for a bacterial consortium to learn and classify a pattern without major human intervention at the intermediate steps.

Indeed the architecture and algorithm are inspired by *in silico* operations. But translating the operations from simulations to real living cells is not a straightforward task. When implementing computations in living cells, we face challenges such as genetic circuit noise, cross-talks, leaky basal expression and so on. In our study, we designed circuit parts to manage cross-talks, and fine-tuned the transfer function of receiver circuit to optimize decision-making. In particular, we adjusted the parameters in protocols to align the output dynamic range of senders to the input operation range of receiver. This is an important step to scale-up the genetic perceptron to multi-layer networks.

Also, the algorithm we used is not a standard ANN algorithm, but one with constrained parameters, since the weights are not strictly in continuous levels. We modeled the sender circuits, as well as the activation function in receivers, using biologically relevant equations, rather than taking the conventional formula from machine learning. This allows our approach to be more relevant to design biological systems.

Furthermore, there can be multiple ways to implement weights. Our method demonstrates one feasible implementation using mutated and customized promoters. One special feature in our approach is the negative weight implemented using a repressor promoter. It provides a transfer function with a downward slope, which enables a weight of negative sign and this is not immediately available to implementations using substrate concentrations. As in Fig S19, we illustrated that negative weight values are essential to classify certain patterns.

We agree with the reviewer that developing *in vivo* training would be far more interesting. In future works, we will design and build systems capable of self-adaptive *in vivo* learning, possibly using CRISPR related techniques such as base editing [4].

My main technical concern is about the use of aTc in “zero-bit” wells. As authors explain, its purpose is to avoid cross-talk between different sender colonies via residual OC6 after they are mixed together in the receiver well, because aTc would repress gene expression in the colonies which were exposed to it. Aside from the issue that it significantly diminishes the generality of the proposed classification scheme (we have to know a priori where there are “zeros” to add aTc there), I don’t understand how this addresses the issue of cross-talk.

The cross-talk could arise due to senders at “0” states being accidentally activated by residual OC6 from senders at “1” states. We performed an assay as in Fig S13 to demonstrate that cross-talks lead to distorted output in receivers and pre-incubation with

aTc reduces such distortions.

From a logic circuit design perspective, the aTc controlled component in senders resembles an auxiliary unit in electronic circuits, the state of which can be programmed externally. Auxiliary units are common in engineering, especially when implementing logic functions. Alternatively, we can also implement sender circuits using light-sensitive components. In this way, sender activities can be turned off instantaneously without residual effects to cause cross-talks.

A more favorable approach to prevent cross-talks in senders is to use microfluidic devices. Here we demonstrated the idea by using a simple fluidics device, using which we can achieve recognition of 2-bit patterns without adding aTc (Fig S7-9). In the revised manuscript, we also implemented an alternative design by increasing the transfer function thresholds of sender circuits. Thus, the residual OC6 would not be sufficient to activate other senders. As shown in Fig S10a, sender circuits with positive feedback regulations exhibit steeper transfer functions with higher thresholds than open loop circuits. In this case, simply diluting senders 100 times alone is sufficient to reduce cross-talks from residual OC6 (Fig S10b). The design is also capable of classifying 4-bit patterns (Fig S11).

Indeed, by the same logic, residual aTc would suppress gene expression not only in the well in which it was added, but in other senders as well, after they are mixed together.

We agree that hypothetically there are potential effects of aTc on other senders and illustrated the effects in Fig S12a. Senders with P_{lux} activators (the orange curve) and P_{lux} repressors (the blue curve) are incubated with OC6 first and then diluted in exposure to aTc. The potential effects of aTc are twofold: aTc can reduce the activities of P_{lux} activator senders at “1” states, and may reactivate P_{lux} repressor senders that have already become inactivated.

The former effect is not a critical issue because lactones including OHC14 are highly stable [5]. Given sufficient incubation time, these P_{lux} activator senders can produce enough signaling molecules OHC14 to activate receivers. The latter reactivation effect, however, may distort classification results. To examine this type of cross-talk, we measured the time course responses for the previously inactivated P_{lux} repressor senders in exposure to varying aTc concentrations, observing their reactivation. As shown in Fig S12b, we noticed an increase in mCherry levels around 150 min. Thus, we estimated an incubation period of up to two hours before observing inactivated senders become reactivated. In our protocol, senders were mixed and incubated with receivers for 90 min, which is within the two-hour safe period. Furthermore, we also performed an assay to experimentally verify that following our protocol, there was no “reactivation” observed in P_{lux} repressor senders in exposure to aTc (Fig S12c). The above evidence indicates that following our protocol, the hypothetical effects of aTc do not distort the classification

performance.

Furthermore, it seems that there is a much simpler way to eliminate this kind of cross-talk, namely, separate receiver cells from their media after cultivation in the presence of OC6, and only use media from different wells, not cells.

We agree with the reviewer that there can be alternative methods to prevent cross-talks. As discussed in the previous comment, reactions can be implemented in a fluidics device that allows only signaling molecules to diffuse across a channel. Thus, cross-talks can be avoided without using aTc. In future works, we expect to design devices for more sophisticated patterns. For now, our study serves as a proof-of-principle that algorithms for *in silico* devices can be realized in living cells. We also thank the reviewer for your suggestion on separating cells from their media, and will test it in future works.

Overall, I think the paper lacks sufficient conceptual novelty that could make it suitable for Nature Communications. After an appropriate revision, it can probably be published in a more specialized journal, such as ACS Synthetic Biology.

We still thank the reviewers for your comments.

References

1. Baldassi, C., Braunstein, A., Brunel, N. & Zecchina, R. Efficient supervised learning in networks with binary synapses. *Proceedings of the National Academy of Sciences* **104**, 11079–11084. ISSN: 0027-8424, 1091-6490 (June 2007).
2. Hubara, I., Soudry, D. & Yaniv, R. E. Binarized Neural Networks. *30th Conference on Neural Information Processing Systems*. arXiv: 1602.02505 [cs.LG] (2016).
3. Meyer, A. J., Segall-Shapiro, T. H., Glassey, E., Zhang, J. & Voigt, C. A. Escherichia coli “Marionette” strains with 12 highly optimized small-molecule sensors. *Nature Chemical Biology* **15**, 196–204. ISSN: 1552-4450, 1552-4469 (Feb. 2019).
4. Farzadfard, F. *et al.* Single-Nucleotide-Resolution Computing and Memory in Living Cells. *Molecular Cell* **75**, 769–780.e4. ISSN: 10972765 (Aug. 2019).
5. Weber, M. & Buceta, J. Noise regulation by quorum sensing in low mRNA copy number systems. *BMC Systems Biology* **5**, 11. ISSN: 1752-0509 (Dec. 2011).
6. Hooshangi, S., Thiberge, S. & Weiss, R. Ultrasensitivity and noise propagation in a synthetic transcriptional cascade. *Proceedings of the National Academy of Sciences* **102**, 3581–3586. ISSN: 0027-8424, 1091-6490 (Mar. 2005).
7. Rizik, L., Ram, Y. & Daniel, R. Noise Tolerance Analysis for Reliable Analog and Digital Computation in Living Cells. *Journal of Bioengineering & Biomedical Science* **06**. ISSN: 21559538 (2016).
8. MacKay, D. J. C. *Information theory, inference, and learning algorithms* ISBN: 9780521642989 (Cambridge University Press, 2003).
9. Gardner, E. Maximum Storage Capacity in Neural Networks. *Europhysics Letters (EPL)* **4**, 481–485. ISSN: 0295-5075, 1286-4854 (Aug. 1987).
10. Krauth, W. & Mézard, M. Storage capacity of memory networks with binary couplings. *Journal de Physique* **50**, 3057–3066. ISSN: 0302-0738 (1989).
11. Tamsir, A., Tabor, J. J. & Voigt, C. A. Robust multicellular computing using genetically encoded NOR gates and chemical ‘wires’. *Nature* **469**, 212–215 (Jan. 2011).
12. Kyllilis, N., Tuza, Z. A., Stan, G.-B. & Polizzi, K. M. Tools for engineering coordinated system behaviour in synthetic microbial consortia. *Nature Communications* **9**, 2677. ISSN: 2041-1723 (Dec. 2018).
13. Gurbatri, C. R. *et al.* Engineered probiotics for local tumor delivery of checkpoint blockade nanobodies. *Science Translational Medicine* **12**, eaax0876. ISSN: 1946-6234, 1946-6242 (Feb. 2020).

14. Toda, S., Blauch, L. R., Tang, S. K. Y., Morsut, L. & Lim, W. A. Programming self-organizing multicellular structures with synthetic cell-cell signaling. *Science*. ISSN: 0036-8075, 1095-9203.

Reviewers' Comments:

Reviewer #1:

Remarks to the Author:

The authors have addressed all my suggested edits. I suggest acceptance.

Reviewer #2:

Remarks to the Author:

The authors have done substantial revisions considering the referees' comments. There is now an impressive number of 31 supplementary figures (6 in the first version) and 10 supplementary notes (3 in the first version).

I particularly appreciate the additional sections that directly address the points I raised: Cross-talk discussion, Alternative implementations, Negative weight, Capacity of perceptron, and scale-up to multi-layer networks.

Regarding point number 4, Fig. S13 is nice but I was not challenging the fact that cross-talk occurs but rather how to prevent it. Anyhow this is now covered in an additional supplementary section.

On point 5, the two options to implement negative weights are important and would deserve to be mentioned in the main text. Also, there are other quenchers for OHC14 this should be discussed in the Negative weight section.

Related to point 6, even prior to measuring fluorescence the 4 missing patterns have $w \times p$ values in the boundary range [4000, 4500]. Perhaps, this observation could be added to the text. Could the range [4000, 4500] be reduced to allow classification of all patterns of course in agreement with fluorescence levels? What would be that range?

On point 9 (Fourth), negative weights can be implemented in metabolic perceptron adding enzymes taking as substrate(s) the product(s) of the enzymes corresponding to the positive weights.

Reviewer #3:

Remarks to the Author:

Authors substantially revised their manuscript and improved it in many respects. I appreciate that they have demonstrated that adding aTc to "zero" chambers does not lead to reactivation of repressor promoters within a certain time window.

However, I still have lingering doubts regarding the significance of the obtained results. In my main comment I noted that their perceptron was in fact a multi-step process with many manual interventions: after input OC6 signals are added to wells with "sender" cells and cultivated, the sender cells with their media containing various amounts of OC14 molecules, needed to be manually collected, diluted, mixed together and placed in a different chamber with quorum-sensing "receiver" cells that in turn would produce the output signal (GFP). To show that a more "integrated" procedure is possible, authors developed and very briefly described in the Discussion a microfluidic system with three wells connected near the bottom by a 100 μ m-deep channel that supposedly should pass quorum-sensing signal (OC14) among the wells. Wells 1 and 3 were for sender cells, whereas the middle well 2 was for the receiver cells. While this system is certainly a step in the right direction, the description of experiments with this new system in Figs. S7-S9 is rather sketchy and leaves more questions than it answers, however. First of all, how could cells in the three wells be kept separate if the height of the channel (100 μ m) was so much bigger than cell diameter (1 μ m). Second, the functionality of the classifier was not really demonstrated, since only

a single (1,1) pattern, high OC6 in both well 1 and well 3, was presented. We cannot be sure that the system actually works as a classifier unless other patterns (0,0), (1,0), (0,1) are also presented and yield separable output signals in well 2. Third, it is not clear why wouldn't input OC6 flow through the channel just as well as OC14, and so why would not cells in wells 1 and 3 sense some weighted mixtures of input signals. I think this new potentially interesting addition had to be done seriously to be presented as a part of the paper, and not just as an afterthought, as it looks now.

Synthetic neural-like computing in microbial consortia for pattern recognition

Response to Reviewers Letter

We thank all reviewers for the comments. We included new supplementary figures and displayed them in the response letter for convenience. **Our answers to reviewers' comments are in bold.**

REVIEWER COMMENTS

Reviewer #1 (Remarks to the Author):

The authors have addressed all my suggested edits. I suggest acceptance.

We thank the reviewer for appreciating our study.

Reviewer #2 (Remarks to the Author):

The authors have done substantial revisions considering the referees' comments. There is now an impressive number of 31 supplementary figures (6 in the first version) and 10 supplementary notes (3 in the first version).

1. I particularly appreciate the additional sections that directly address the points I raised: Cross-talk discussion, Alternative implementations, Negative weight, Capacity of perceptron, and scale-up to multi-layer networks.

We thank the reviewer for the comment.

2. Regarding point number 4, Fig. S13 is nice but I was not challenging the fact that cross-talk occurs but rather how to prevent it. Anyhow this is now covered in an additional supplementary section.

We are glad that our response answered the reviewer's question.

3. On point 5, the two options to implement negative weights are important and would deserve to be mentioned in the main text. Also, there are other quenchers for OHC14 this should be discussed in the Negative weight section.

We agree with the reviewer. We added the note on two possible ways to implement negative weights in the main text (page 5, lines 162-166). We also noted in the main text and the Negative weight section regarding lactone quenchers other than AiiA. Since many quorum quenchers can act on a range of lactones, adopting them for circuit designs requires careful characterization of their specificities.

4. Related to point 6, even prior to measuring fluorescence the 4 missing patterns have $w \times p$ values in the boundary range [4000, 4500]. Perhaps, this observation could be added to the text. Could the range [4000, 4500] be reduced to allow classification of all patterns of course in agreement with fluorescence levels? What would be that range?

We added a note regarding the boundary range into the main text (page 6, lines 183-187).

Hypothetically, we can adjust the receiver transfer function to change the boundary range and improve pattern classification. As a demonstration, we simulated the classification performance using a transfer function with an increased hill-coefficient “ n ” following the equation as below.

$$f(x) = \beta_m \frac{(x/K_d)^n}{1 + (x/K_d)^n} + \beta_0 \beta_m$$

Here K depends on the binding affinity between the promoter and inducer-transcription factor (TF) complex. n describes the cooperativity of binding. As shown in Fig S16 below, increasing n from 2.33 to 4.0 leads to a sharpened receiver transfer function (Fig S16a). Here the parameter n that equals 2.33 is obtained by fitting experimentally measured mCherry levels in senders and the corresponding EYFP levels in receivers (Fig 4c). With n equals 4, we improved classification results (Fig S16b), as the four patterns within the boundary zone (grey bars) become more separate from the patterns in the “high” category (six blue bars on the right).

Notably, n equals 4 is much higher than the cooperativity involved in synthetic biomolecular interactions. Nevertheless, one possible way to achieve ultra-sensitivity is to cascade multiple layers of genetic circuits (Hooshangi et al., 2005). We demonstrated this mechanism by simulating the transfer function of a two-layer genetic circuit (Fig S16c). The first layer contains a promoter that drives a gene to produce TF that activates the promoter on the second layer. In both layers, n and K are kept unchanged as 2.33 and 2820 respectively and both values are biologically realistic. This combination of parameters results in an overall transfer function that approximates the target function with the desired n value of 4. We added the above discussion in the Supplementary Notes section Explanation on classifying 4-bit patterns.

For the arbitrarily selected weight set [450, 3500, 900, 3500], the products of weight and 16 patterns are [0, 3500, 900, 3500, 450, 1350, 4400, 4400, 3950, 3950, 7000, 7900, 4850, 7450, 4850, 8350]. Values for the four patterns within the boundary are italicized and underlined. When n equals 2.33, the corresponding network output calculated as in Fig 4a is [0.02, 0.39, 0.04, 0.39, 0.02, 0.08, 0.52, 0.52, 0.46, 0.46, 0.77, 0.82, 0.57, 0.79, 0.57, 0.84]. In this case, [4000, 4500] that results in output values [0.46, 0.53] was intuitively selected as a boundary range to partition

the 16 patterns into two equal sets. With n equals 4, the network output becomes [0.02, 0.3, 0.02, 0.3, 0.02, 0.03, 0.52, 0.52, 0.41, 0.41, 0.88, 0.93, 0.61, 0.91, 0.61, 0.95]. The same output values [0.46, 0.53] corresponds to [4150, 4450] in sender fluorescence, which is narrower than [4000,4500].

Fig S16 **a**. Classification outcome can be adjusted by tuning the receiver transfer function. Based on simulations, the receiver transfer function becomes sharpened as n increases from 2.33 to 4. Here the black curve is fitted using the Hill equation to experimental measurement as in Fig 4c with n equals 2.33. **b**. Simulated classification of the 16 patterns is improved as n increases from 2.33 to 4. (Upper left panel) When n equals 2.33, the bars in grey color are in an intermediate

category and cannot be readily classified into either “high” or “low” patterns. (Lower right panel) When n equals 4, the grey bars are more separate from the “high” category and can be classified as “low”. **c.** Ultra-sensitivity in receiver transfer function can be achieved by cascading two layers of genetic circuits. The curves are simulated results. **d.** Experimentally measured receiver output presented in median EYFP levels for the complete 16 patterns. The blue bars correspond to the 12 patterns shown in Fig 3c. The additional 4 patterns are in grey.

5. On point 9 (Fourth), negative weights can be implemented in metabolic perceptron adding enzymes taking as substrate(s) the product(s) of the enzymes corresponding to the positive weights.

We thank the reviewer for the suggestion of using enzymes to implement negative weights. We will look into it for designs in future works.

Reviewer #3 (Remarks to the Author):

Authors substantially revised their manuscript and improved it in many respects. I appreciate that they have demonstrated that adding aTc to “zero” chambers does not lead to reactivation of repressor promoters within a certain time window.

We thank the reviewer for the comment. Indeed timing is important in our protocol. We highlight this idea again in the new set of experiments using fluidic devices.

1. However, I still have lingering doubts regarding the significance of the obtained results. In my main comment I noted that their perceptron was in fact a multi-step process with many manual interventions: after input OC6 signals are added to wells with “sender” cells and cultivated, the sender cells with their media containing various amounts of OC14 molecules, needed to be manually collected, diluted, mixed together and placed in a different chamber with quorum-sensing “receiver” cells that in turn would produce the output signal (GFP).

We agree with the reviewer. We have shown that the framework can be adapted alternatively by using more than one type of chemical species as inputs. In this case, all reactions occur in “one-pot” without many manual steps (Fig S10).

a

b

Fig S10 Sender circuits with positive feedback regulation. **a.** (Upper) Genetic circuit diagram with positive feedback (PF) design. (Lower) Transfer functions of the PF circuits for three mutated P_{lux} promoters. **b.** Patterns used in an assay to demonstrate that no cross-talks between senders have been observed. “m8” stands for P_{lux} mut8, a mutated P_{lux} promoter with weak strength. “m40” stands for P_{lux} mut40, a mutated P_{lux} promoter with high strength. All three patterns correspond to the sender mix that are supposed to result in low activation levels in receivers. In all three patterns, high concentrations of inducer OC6 were applied to P_{lux} mut8 only. If cross-talks occur, P_{lux} mut40 could become affected and produce OHC14 to activate receivers. As shown from the right figure, the median levels of receivers corresponding to the three patterns are consistently below 1000 a.u., indicating low activation levels in receivers.

2. To show that a more “integrated” procedure is possible, authors developed and very briefly described in the Discussion a microfluidic system with three wells connected near the bottom by a 100um-deep channel that supposedly should pass quorum-sensing signal (OC14) among the wells. Wells 1 and 3 were for sender cells, whereas the middle well 2 was for the receiver cells. While this system is certainly a step in the right direction, the description of experiments with this new system in Figs. S7-S9 is rather sketchy and leaves more questions than it answers, however.

We are glad that the reviewer agrees with us on the importance of microfluidic systems. In this revision, we have added a new section in Supplementary Notes section Fluidics experiments to describe the detailed protocol.

2.1. First of all, how could cells in the three wells be kept separate if the height of the channel (100um) was so much bigger than cell diameter (1um).

We appreciate the reviewer for the insight. Indeed bacteria cells can enter the channel and freely diffuse in it. However, within the time frame as in our protocol, sender bacteria inside wells are most likely still in the channel (please see the Supplementary videos). We demonstrate this scenario experimentally by observing the movement of bacteria using a microscope (Nikon Ti). We filled three wells with LB media and added 1 μ l of bacteria culture grown overnight to one well on the side (either well1 or well3). We used a microscope camera (Andor Neo SCC-02287) to capture a view close to the channel every ten minutes (more detail is available Supplementary Notes section Fluidics experiments). We presented two time-lapse recordings as Supplementary videos to show the movement of bacteria.

2.2. Second, the functionality of the classifier was not really demonstrated, since only a single (1,1) pattern, high OC6 in both well 1 and well 3, was presented. We cannot be sure that the system actually works as a classifier unless other patterns (0,0), (1,0), (0,1) are also presented and yield separable output signals in well 2.

We thank the reviewer for the suggestion. In the revised version, we fabricated new devices and performed more experiments for all four patterns. We used

wild-type P_{lux} for both weights in well1 and 3. This arrangement gave rise to an “AND” function. As expected, only pattern (1,1) leads to high activation in receivers (Fig S8, the figure below). We measured from three devices and presented the median values of fluorescence levels averaged from these devices.

Fig S8. The three-well fluidic devices can classify 2-bit patterns. Experiments performed using equal weights (wild-type P_{lux}), effectively acting as an “AND” classifier. (Upper panel) GFP levels in well2 for the four 2-bit patterns. (Lower panel) Median mCherry levels in well1 and well3. Experiments were performed on three devices. Devices were autoclaved to eliminate contaminants after each experiment. More experimental detail is available in Supplementary Notes section Fluidics experiments.

2.3. Third, it is not clear why wouldn't input OC6 flow through the channel just as well as OC14, and so why would not cells in wells 1 and 3 sense some weighted mixtures of input signals.

We appreciate the reviewer for the comment. Indeed both the inducer OC6 and the signaling molecule OHC14 can diffuse freely in the device. Both molecules travel from the side wells to the center within a comparable time frame. OHC14 molecules in well1 or 3 need to travel the distance between a side well and the center to activate receivers in well2. In contrast, OC6 molecules need to diffuse more than

that distance (e.g., from one side well to another) to interfere with senders in the other well. Intuitively, it takes more time for OC6 in well1 to react with senders in well3. As a demonstration, using plate-reader we measured time courses of mCherry levels from three wells in response to OC6 added to well1. In this setup, sender cells were added to well2 and 3. OC6 molecules in well1 diffuse across the channel to activate senders in well2 and 3. As in Fig S9, the activation of senders in well3 started around 5 hours. Even if we take into account that cells in well3 move along the channel, it takes time for OC6 to react with these senders and produce OHC14. These steps lead to a further delay in subsequent activation of receivers by the resultant OHC14.

Thus, the distinction in travel distance between the two chemicals OC6 and OHC14 reduces the concern that OC6 might interfere with other side wells to cause “weighted mixtures of input signals”. This notion has also been supported by FACS measurement. From FACS experiments (Fig S8), we did not observe the activation of senders in wells without OC6 inputs. Nor did we observed GFP levels in receivers being affected by senders with “0” inputs since both patterns “01” and “10” result in low GFP in receivers.

Fig S9 (Upper) Experiment setup for the assay measured in plate-reader. The device was fixed on a plate-reader holder for measurement. OC6 (final concentration 10 μ M) was added to well1. Senders (1 μ l) from overnight bacteria solution were added to well2 and well3. Time courses of mCherry levels in three wells normalized to OD. Since well1 contains no bacteria, the mCherry level in well1 stays low and is presented as a control reference.

3. I think this new potentially interesting addition had to be done seriously to be presented as a part of the paper, and not just as an afterthought, as it looks now.

We are grateful for the overall feedback from the reviewer. We emphasized the fluidics design in the main text (page 4, lines 110-116) and presented further details regarding device implementation in the Supplementary Notes section Fluidics experiments. In the revised manuscript, we have fabricated new devices and performed repeated experiments.

We have devoted much effort to present the microfluidic system and experimental results. We showed that the approach based on cell consortia works well as a design framework to implement perceptron networks for chemical pattern recognition. We will devote more effort in future works to address specifics regarding design and implementations using microfluidic devices.

Reviewers' Comments:

Reviewer #2:

Remarks to the Author:

The authors have addressed all the points I raised in my previous review, I suggest acceptance of this last version

Reviewer #3:

Remarks to the Author:

I think the authors did a good job addressing my and the other reviewer's remarks in their second revision. I think the manuscript can be published in its present form.